# Age- and Stage-Dependent Prostate Cancer Aggressiveness Associated with Differential Notch Signaling

**DOI:** 10.3390/ijms24010164

**Published:** 2022-12-22

**Authors:** Magdalena Julita Orzechowska, Dorota Anusewicz, Andrzej K. Bednarek

**Affiliations:** Department of Molecular Carcinogenesis, Medical University of Lodz, 90-752 Lodz, Poland

**Keywords:** prostate cancer, age, Notch signaling, disease progression, cytoskeleton reorganization, epithelial-to-mesenchymal transition, hybrid epithelial/mesenchymal state, tumor invasiveness, remodeling of the tissue architecture

## Abstract

Prostate cancer (PC) remains a worldwide challenge, as does the question of how to distinguish its indolent from its aggressive form to reconcile proper management of the disease with age-related life expectations. This study aimed to differentiate the Notch-driven course of PC regarding patients’ ages and stage of their disease. We analyzed 397 PC samples split into age subgroups of ≦55, 60–70, and >70 years old, as well as early vs. late stage. The clinical association of Notch signaling was evaluated by DFS and UpSet analyses. The clustering of downstream effectors was performed with ExpressCluster. Finally, for the most relevant findings, functional networks were constructed with MCODE and stringApp. The results have been validated with an independent cohort. We identified specific patterns of Notch expression associated with unfavorable outcomes, which were reflected by entering into a hybrid epithelial/mesenchymal state and thus reaching tumor plasticity with its all consequences. We characterized the molecular determinants of the age-related clinical behavior of prostate tumors that stem from different invasive properties depending on the route of the EMT program. Of the utmost relevance is the discovery of age- and stage-specific combinations of the Notch molecules predicting unfavorable outcomes and constituting a new prognostic and therapeutic approach for PCs.

## 1. Introduction

Prostate cancer (PC) is the most frequently diagnosed cancer type in men and leads the most cancer-related mortality rates right after lung cancer, globally [1]. Elderly men comprise the majority of PC cases; however, recently, an increasing trend is reported among young males [2]. The course of PC is very heterogeneous, posing major challenges in proper management of the disease and accurate risk stratification. The former is strongly related to the age of PC onset and requires individualized decisions determined by balancing the benefits and risks of active treatment while considering comorbid conditions, life expectancy, and tumor characteristics [3]. In turn, risk stratification aims to distinguish indolent from aggressive PCs of the advanced stage. The latter is much more frequently diagnosed among younger males and reaches the highest mortality among all age groups, hence early-onset PC having been already determined to be a distinct clinical–pathological phenotype with a poor prognosis [4]. In contrast, older men tend to be diagnosed with tumors of a more advanced stage, but, simultaneously, they also appear to die from other comorbid causes. Many reports show that the rate of disease is associated with increasing age, as a small microscopic focus, and with reduced aggressiveness [5]. The difficulties in adequate PC management to either reduce unnecessary overtreatment of more benign cases or apply efficient strategies for metastatic disease prompt an expansion of the current knowledge, with molecular mechanisms and hallmarks reflecting the age-specific course of prostate tumorigenesis ultimately aiding the stratification of high- from low-risk tumors. Such distinction lays in functional alterations of regulatory pathways driving the cancer cell behavior that is manifested in the clinical observations among individuals. Currently, the early detection of PC, as well as clinical decisions on the management, are mostly based on prostate-specific antigen (PSA) screening. However, PSA is rather a prostate- than prostate cancer-specific biomarker; thereby, the information provided by its measurement is often ambiguous. On the other hand, PSA has been considered a more accurate secondary indicator to monitor PC progression among already diagnosed PC patients [6].

Some of the pathways carry external signals from the environment or neighboring cells, others trigger the internal programs of cellular death and proliferation. Notch is one of the major developmental pathways orchestrating the whole cellular machinery during pre- and postnatal life: proliferation, differentiation, and survival, to determine the fate of the cells and tissue architecture [7]. Unsurprisingly, due to its roles, an overwhelming body of evidence has been collected on cancerogenic consequences of Notch deregulation [8], including on PC [9]. To date, expression profiling studies reveal that the core components of Notch signaling may distinguish aggressive PCs of high Gleason grade and indicate that the pathway is functional especially in progressing metastatic disease [10]. Although most of the Notch-dedicated studies consider the tumorigenic effects of the mutations occurring within the pathway or single-gene alterations, the aberrations of core signaling resulting in cancer progression due to affected downstream output without overt mutations in the core members of the Notch pathway seem of the greatest interest.

In this study, we attempted to differentiate the Notch-driven course of tumorigenesis and disease progression of PC regarding the age of onset as well as the stage of the disease. We aimed to construct a global view of Notch-related expression alterations that occur in the core components, but, more importantly, we also took the downstream consequences of the primary Notch deregulation into consideration to reveal the complex background of age-associated differences in PC course. Finally, we stressed the potential of Notch profiling exploitation in stratifying patients who truly need active treatment compared to those who can remain under surveillance.

## 2. Results

### 2.1. Decreased Notch Activity Restrains PC Progression in Older Patients

The expression profiling of the core Notch components revealed differences between the age groups of the same tumor stage as well as within the same age groups between the tumor stages (Figure 1). The age group of above 70 years old was distinct from the age groups of below 55 or 60–70 years old, which were similar to each other to some extent, especially regarding the late stage of the disease. A more detailed comparison of the Notch expression profiles between the tumor stages revealed the spectrum of Notch activity reaching a peak at 60–70 years old. At the early stage of PC among patients, the age group of under 55 years old expressed ligands (*DLL1*, *DLL3*, *JAG1*), Notch-specific TFs (*HES1* and *HEY1*), and major components of the γ-secretase complex (*PSEN1*, *PSEN2*, *PSENEN*, *NCSTN*). We also observed increased activity of epigenetic regulators such as histone deacetylases (*HDAC1*, *HDAC2*), as well as cross-talk with the Wnt pathway through *DVL3*. At the late stage of PC in patients below 55 years of age, similarly to early and then late PC in 60–70-year-olds, we observed heightening of Notch activity through triggering the expression of additional ligands (*DLL4*, *JAG2*, *DLK1*), Notch receptors, TFs (*HES4*, *HES5*, *HEY2*, *HEYL*, *PTCRA*), and intensified cross-talk with the Wnt pathway through DVLs (*DVL1*, *DVL2*) until its activity extinction in elderly men, especially at the late stage of PC. However, we also observed disparate epigenetic regulation through histone deacetylases (*HDAC1*, *HDAC2)*, *CREBBP*, and *EP300*, which switched on and off in different PC subgroups, accordingly (Table 1). Detailed results of the comparison may be found in the Appendix A and GitHub repository (https://github.com/orzechmag/notch-age-pc).

Moreover, the core Notch signaling demonstrated the specific age- and tumor stage-related expression signatures enclosed in Appendix A. We identified a unique expression profile of 13 Notch genes among patients below 55 years old, comprising, among others, *DLL3* and *HEY1* at the early stage, whereas *JAG2*, *DLL4*, *HEYL*, *NOTCH3*, *NOTCH4*, and *DVL1* were relevant at the late stage. Furthermore, 18 genes were characterizing tumors of 60–70-year-olds, such as *HES1*, *HES5*, and *HEY2* at the early stage and *NOTCH1*, *JAG1*, *RBPJ*, *NCSTN* at the late stage. Finally, 37 genes were unique for tumors in elderly men, including *JAG1*, *DLL4*, *DVL1*, *NOTCH1*, *NOTCH2*, *NOTCH3*, *PTCRA*, *NCSTN*, and *RBPJ* at the early stage, while *DLL3*, *DLK1* were relevant for the late stage of the disease. Significantly, 13 genes overlapped between groups of patients below 55 and 60–70 years old (early vs. late tumor stage DOWN-UP: *KAT2B*, *NUMB*, *DVL2*, *MAML1*, *PTCRA*, *CREBBP*, *MAML2*, *NOTCH2*, *APH1B*, *ADAM17*, *RBPJL*, UP-DOWN: *SNW1*, *LFNG*), while none of the genes intersected between these groups and the elderly men. The summary of the above comparisons is shown in Table 2.

### 2.2. The Specific Combination of Jagged Ligands and Notch Receptors Drives the Metastatic Potential of PC

We aimed to evaluate the effects of the particular Notch members profiles on DFS analysis among age groups, and stage of the disease, accordingly. Detailed results are shown in Table 3. Among the major findings, we observed contrasting effects of *JAG1* and *JAG2* on DFS with increasing age, although independently of the disease stage (*JAG2:* HR_≦55_ = 3.03, *p* = 0.028; HR_60–70_ = 6.12, *p* = 0.044; late-stage HR = 1.94, *p* = 0.043; HR_>70_ = 0.069, *p* = 0.012; *JAG1*: HR_>70_ = 0.079, *p* = 0.007; late-stage HR_>70_ = 0.093, *p* = 0.012). The same trend could be also observed among Notch receptors: *NOTCH4* significantly altered the prognosis among the group below 55 years old (HR = 3.34, *p* = 0.023) and in patients above 70 years old (HR = 0.069, *p* = 0.012). In turn, *NOTCH2* and *NOTCH3* were associated with DFS in patients aged 60–70 years old (*NOTCH2*: HR = 1.89, *p* = 0.041; *NOTCH3*: HR = 2.37, *p* = 0.007; early-stage HR = 24.9, *p* = 0.0002), but they were not significant in the other age or stage groups. Furthermore, some of the Notch genes differentiated DFS with tumor stage within patients aged 60–70 years old, such as *ADAM10*, *DTX3L*, *DVL3*, and *HDAC1,* which were significantly lowered at the early stage (*ADAM10*: HR = 14.4, *p* = 0.0043; *DTX3L*: HR = 11.2, *p* = 0.016; *DVL3*: HR = 9.73, *p* = 0.023; *HDAC1*: HR = 9.4, *p* = 0.025) and heightened at the late stage (*ADAM10*: HR = 0.373, *p* = 0.011; *DTX3L*: HR = 0.515, *p* = 0.034; *DVL3*: HR = 0.299, *p* = 0.0019; *HDAC1*: HR = 0.398, *p* = 0.037) of PC. We also performed the UpSet analysis (form of Venn diagrams) to determine whether some of the PC patients bear a specific Notch signature, i.e., the intersect of the Notch core members profiles, that is associated with DFS prognosis (Appendix A). Among 53 patients below 55 years old at the early stage of the disease, we identified a subgroup of 20 patients (37.7%) with a Notch signature consisting of *RBPJ*, *ADAM17*, *RBPJL*, and *ATXN1*; however, there was no significant difference in DFS between the possessors and non-possessors of the signature (HR < 0.001, *p* = 0.23, Figure 2). At the late stage of the disease, half of the group aged below 55 years old showed the signature composed of *KAT2A*, *LFNG*, *HDAC2*, *SNW1*, and *PSEN2* corresponding to significantly better DFS rates (HR = 0.081, *p* = 0.0028, Figure 2). In the group of PC patients aged 60–70 years old at the early stage of the disease, 58 patients (69%) had a common Notch profile of *NOTCH3*, *APH1A*, *HDAC2*, *MFNG*, and *DTX4,* improving the DFS prognosis (HR < 0.001, *p* = 0.021, Figure 2), while 124 (73.4%) patients at the late stage of PC showed the profile of *HES5*, *ADAM17*, *CREBBP*, *DVL3*, and *ADAM10* (HR = 0.35, *p* = 0.0016, Figure 2). Finally, we found a favorable Notch signature among 6 (27.3%) late-stage PC patients aged above 70 years old, consisting of *JAG1*, *PTCRA*, *HEYL*, *DTX3*, and *RFNG* (HR < 0.001, *p* = 0.013, Figure 2). The detailed statistics of the DFS analysis for carriers and non-carriers of the identified Notch signature are shown in Table 4.

### 2.3. Notch Signaling Targets Tumor Properties Such as Aggressiveness and Invasive Potential

Alterations within the core Notch signaling entail deregulation of the downstream biological processes executed by the targets of Notch-specific TFs, such as the HES and HEY families manifesting in the clinical characteristics of the tumor. By the clustering of the Notch effectors, we identified 188 unique profiles therein, within which we determined specific clusters of high biological relevance, distinctive between the age groups and stage of the disease (Table 5, Appendix A). Then, we constructed a functional network of high confidence subclusters based on the topology to find densely interconnected regions representing intrinsic/entangled pathways or processes. It led us to unravel that Notch signaling targets PC properties such as aggressiveness and invasive potential, which are modified along with progressing age and stage of the disease. Though the genes predominating within the network were associated mainly with the cytoskeleton (actin microfilaments, intermediate filaments, microtubules, and molecular motors), adhesion, extracellular matrix (ECM), establishing cell polarity, histone modifications, and epithelial-to-mesenchymal transition (EMT)-specific transcription factors, as well as angiogenesis, apoptosis, and cell cycle, our investigations revealed four major subclusters of Notch—EMT regulatory loop, kinesin motors, integrin signaling, and cellular migration—as the highly interconnected players driving PC progression (Figure 3).

#### 2.3.1. Reshaping of Intercellular Communication—Cytoskeleton, Basement Membrane, and Extracellular Matrix

Alterations within the architecture of actin microfilaments were mostly specific for the age group below 55 years old and concurrently contrasted to mutually similar groups of 60–70 and above 70 years old. In addition, these profiles reverted at the late stage vs. early stage of PC. In turn, the profiles of the components of the intermediate filaments, such as *FLG* (filaggrin) and *VIM* (vimentin), distinguished the group of 60–70-year-olds from all groups at the early stage of the disease, with lowered *FLG* expression and heightened *VIM* expression therein; whereas, with PC progression, elevated *FLG* expression was kept only among patients above 70 years old, and significantly, this group acquired *VIM* expression while the others maintained the early stage profile. Furthermore, the PC early-stage expression of the microtubules constituents was mostly increased in patients above 70 years old, whereas higher expression of the molecular motors-related genes distinguished younger patients below 55 years old. In contrast, at the late stage of PC, the microtubules and molecular motors genes showed unified patterns of expression. Particularly, patients below 55 years old demonstrated a decrease in expression, while the groups aged 60–70 and above 70 years old presented the rise. Among adhesion-related molecules, the expression of *CDH1* (E-cadherin) was higher among patients below 55 years and above 70 years at the early stage of the disease and, significantly, the *CDH1* expression was lost among patients below 55 years old with stage progression. In turn, *CTNNB1* (β-catenin) and *CTNNA2* (α-catenin) showed opposite profiles differentiating patients at both stages of the disease, although shifting in the patterns. Specifically, we observed an increase in the *CTNNB1* expression among patients below 55 years, in contrast to the groups of 60–70 and above 70 years old, while *CTNNA2* was decreased in the former and elevated in the latter at the early stage of PC. With disease progression, *CTNNB1* and *CTNNA2* switched their expression patterns to the opposite in comparison to the early PC stage (*CTNNB1*: downregulation in the group below 55 years old vs. upregulation in the groups of 60–70 and above 70 years old; *CTNNA2*: upregulation in the groups of below 55 and 60–70 years old vs. downregulation in the group below 55 years old).

Among ECM-related genes, collagens (*COL4A4*, *COL6A2*, *COL8A1*, *COL14A1*), integrins (*ITGA1*, *ITGA6*, *ITGA8*, *ITGA10*, *ITGAV*, *ITGB2*, *ITGB8*), as well as *MMP7*, were mostly overexpressed in patients below 55 years old, rather than in the groups of 60–70 and above 70 years old at the early stage of PC, and the profile switched to the opposite with stage progression.

#### 2.3.2. Complementary Alterations—The Important Players

Of the EMT-specific transcription factors, *SNAI2* maintained its expression profile along with stage progression, although it altered with age: it showed downregulation in patients below 55 years old, whereas it was increased in the groups of 60–70 and above 70 years old. *TWIST1* expression was lowered in the groups below 55 and above 70 years old at the early stage of PC, and the former group additionally gained the *TWIST1* expression at the late stage. *ZEB2,* in turn, demonstrated opposite profiles with the progression of PC, as it was upregulated in patients below 55 years old in contrast to the groups of 60–70 and above 70 years old at the early stage, but, at the late stage, the expression dropped in the former and heightened in the latter groups. Additionally, we observed similar patterns in the expression of antiapoptotic *BCL2*, cell cycle regulator and suppressor *RB1*, epigenetic modificators (*CREBBP*, *EP300*), and hypoxia-related transcription factor *HIF1A*. Finally, we found several other notable alterations of a different character. Particularly, *CDK12* expression was heightened only among patients below 55 years old at the early stage and above 70 years old at the late stage of PC. In turn, *CDKN1A* was lowered among patients above 70 years old at the early stage. In contrast, the pattern switched into overexpression among patients below 55 years old and downregulation in the older groups with PC progression. Interestingly, *TP63* showed the same pattern of expression independently of the disease stage, involving downregulation among patients below 55 years old compared to older patients. Similarly, *COCH* expression was heightened only among patients above 70 years old, independently of disease progression. *ESR1* expression was, in turn, decreased in this group, compared with the younger patients at the early PC stage, although the profile switched to the opposite with stage progression.

#### 2.3.3. Specified Pattern of Notch Signaling Escalates PC Invasiveness through Affecting Plasticity

At the end, we attempted to evaluate the tumor properties associated with aggressiveness and invasive potential among the patients bearing the specific Notch signatures identified in the age and stage subgroups via a functional network consisting of cytoskeleton, adhesion, and ECM-related molecules, as well as key participants of EMT (Figure 4, Table 6). The general finding is that in the PC subgroups with specific Notch signatures associated with unfavorable DFS, these components were upregulated at the early stage of the disease and downregulated at the late stage, unrelated to age. Specifically, at the early stage of the disease, among patients aged below 55 years old that shared the unfavorable profile of the Notch signature (*RBPJ*, *RBPJL*, *ADAM17*, and *ATXN1*), we observed mainly alterations involving the remodeling of cytoskeleton components: actin microfilaments *ACTA2* (actin α2) and *ANXA1* (annexin A1) were downregulated in favor of elevation of *DST* (dystonin), *FLNB* (filamin B), *FN1* (fibronectin 1), *MYO1F* (myosin IF), and *MYO7A* (myosin VIIA); intermediate filaments: *FLG* (filaggrin), cytokeratins *KRT19* and *KRT5,* as well as *VIM* (vimentin); microtubules: tubulins *TUBGCP4* and *TUBGCP5*; molecular motors: dynein *DYNC1H1* and major kinesins *KIF1B*, *KIF21A*, *KIF21B*, *KIF26A*, *KIF2A*, *KIF3A*, *KIF3B*, *KIF5A*, *and KIFAP3.* Regarding adhesion, we observed (high *CDH2* (N-cadherin), *CDH22*, and low *CDH1* (E-cadherin); increases in *CTNNB1* (β-catenin) and *CTNNA2* (α-catenin), *GJA1* (gap junction protein a 1; connexin 43), and *SDC1* (syndecan 1), decreases in *GSC* (goosecoid homeobox), and *MUC1* (mucin 1). Regarding ECM, we observed increases in *COL8A1*, *DDR2* (discoidin domain RTK2), *FBN1* (fibrillin 1), *FRAS1*, integrins such as *ITGA1*, *ITGA6*, *ITGA8*, *ITGAV*, *ITGB2*, and *ITGB6*, laminin *LAMA3*, *LGALS3* (galectin 3), matrix metalloprotease *MMP3*, *NTN4* (netrin 4), and *TNN* (tenascin N), and a decrease in *VTN* (vitronectin), *GDF5*, *ITGA10*, and *ITGA5*. Finally, we identified alterations among the major TFs driving the EMT program, such as a 1.5-fold rise in *SNAI2*, *TCF4*, and *ZEB2* expression, whereas *FOXC2* and *TWIST1* dropped. At the late stage of the disease, among patients bearing the unfavorable profile of the Notch signature (*KAT2A*, *LFNG*, *HDAC2*, *SNW1*, *PSEN2*), we observed that the major components of actin microfilaments were significantly downregulated (*ACTA2*, *ANK2*, *DST*, *FN1*), the same as myosins (*MYO1F* and *MYO7A*) and intermediate filaments including *FLG*, *KRT5*, *KRT19*, and *VIM*. Alterations in adhesion comprised decreases in E-cadherin (*CDH1*) and *CDH11* and a striking drop in *CDH22*. In turn, α- and β-catenins (*CTNNA2*, *CTNNB1*) decreased compared to early-stage PC. On the other hand, we observed diminishing of the expression of ECM molecules (collagens, integrins, laminins, matrix metalloproteases) by at least 1.5 times at the late stage of the disease in patients bearing the unfavorable profile of the Notch signature. Significantly, *ITGA10* and *VTN* were overexpressed, as opposed to their profiles from the early stage of the disease. We also observed noticeable decreases in expression of all major EMT-related TFs, especially *SNAI1* and *SNAI2*. In patients agedbetween 60 and 70 years old at the early stage of the disease, the profiles of diverging subgroups of the DFS-related Notch signature (*NOTCH3*, *APH1A*, *HDAC2*, *MFNG*, *DTX4*) were more diverse than in patients under 50 years old. The components of actin microfilaments were mostly upregulated among patients sharing the unfavorable profile of the Notch signature, but to a lesser extent than in younger patients (e.g., *FLNB*, *FN1*, *MYO7A*). In turn, *ACTA2* showed the same trend towards a decrease in expression. Opposite trends were observed for *ANK2* and *DST*, which were noticeably downregulated, whereas both genes were overexpressed among younger patients (Table 6). Significantly, intermediate filaments (*FLG*, *KRT19*, *KRT5*, *KRT8*) were all downregulated, and, compared to younger patients, they switched to the adversative profiles. Among adhesion molecules, we observed a rise in *CDH1* expression with a simultaneous decrease in the level of *CDH2*. Additionally, *CDH15* and *CDH22* lessened their expression in contrast to younger patients with the unfavorable profile of the Notch signature. Also, *CTNNA2* showed downregulation accompanied by *CTNNB1* elevation. Molecules such as *DSP*, *GSC*, *ICAM1*, *ICAM5*, *MUC1*, *OCLN*, and *TJP1* showed overexpression and reached higher levels than in younger patients, in contrast to *GJA1*, *NID1*, and *SDC1,* which significantly dropped. The expression of the ECM components was more diverse, as some of them gained and some of them lost their expression. In general, collagens showed downregulation, unlike in younger patients. Genes such as *DDR2*, *FBN1*, and *FRAS1* decreased their expression compared to younger patients. Integrins, laminins, matrix metalloproteases, *NTN4*, *TNN*, and *TNXB* decreased their expression, whereas *ITGA10*, *LAMC2*, and *VTN* increased contrastingly to patients under 55 years old. Finally, among TFs, *SNAI2* showed a marked decrease in expression, whereas the remaining ones remained overexpressed. In the late stage of the disease, the carriage of the unfavorable profile of the Notch signature consisting of *HES5*, *ADAM17*, *CREBBP*, *DVL3*, *ADAM10* resulted in downregulation of major adhesion and cytoskeleton-related molecules. Although all actin microfilaments were decreased, *ANK2*, *CALD1*, and *FN1* dropped in the expression, thus resembling the late-stage PC in patients under 55 years old and half-reciprocal to the early-stage PC among 60–70-year-old patients. The intermediate filaments were convergent with both late-stage PC in patients below 55 years old and early-stage PC of 60–70year-old patients, showing downregulation. Besides, microtubules and molecular motors were downregulated and contrasted to early-stage PC among 60–70-year-old patients, as well as late-stage PC in patients below 55 years old. Among adhesion and ECM molecules, we also observed a general decrease in expression, hence demonstrating a unique pattern dissimilar to the comparative age and stage groups. Unlike early-stage PC among 60–70-year-old patients and late-stage PC in patients agedbelow 55 years old, *CDH2* and *CDH4,* as well as *LIN7A*, *COL8A1*, *DSP*, *GSC*, *ICAM1*, *ICAM5*, *FRAS1*, *ITGA10*, *ITGA8*, *ITGAV*, *OCLN*, *PLEC*, *TJP1,* and *NPNT,* showed a drop in expression, whereas *CDH15*, *CDH26*, and *CTNNA2* were overexpressed, akin to late-stage PC in patients below 55 years old. Regarding TFs, only *TWIST1* was increased. Such a pattern was also revealed for late-stage PC in patients below 55 years old. Among patients over 70 years old, it was only possible to analyze the late-stage disease, where we identified the unfavorable Notch signature of *JAG1*, *PTCRA*, *HEYL*, *DTX3*, and *RFNG*. The carriage of the unfavorable profile resulted in lowered expression of actin microfilaments, whereas *MYO1F* was elevated. The same trend was observable in intermediate filaments, especially for *FLG,* as well as for microtubules and molecular motors. In addition, adhesion molecules were lowered, excepting *CDH11*, *DSP*, *GSC*, *MUC1*, *S100A4*, and *SDC1*, similarly to ECM molecules, among which only *ITGB2*, *LAMA1*, *LAMC3*, *MMP9*, and *TNN* showed overexpression. Finally, all TFs were downregulated; however, *TWIST1* was higher, resembling the late-stage PC of the 60–70-year-old patients.

### 2.4. Validation of the Findings

For the cross-validation of our findings, we employed the independent microarray study of Rubicz et al., which resembles the design of our investigations with PC staging (local and regional disease) as well as the patients’ age (≤55, 60–70, and 70> years old). We performed the validation at two levels. Firstly, the comparative profiling of the core Notch members was performed and revealed approx. 47% similarity between the expression patterns in the primary and cross-validation studies. Importantly, we also confirmed shifts in the expression of particular Notch members between disease stages of two (*SNW1*, *DLL3*), three (*PSENEN*, *NUMBL*, *DTX2*), and two (*NCSTN*, *EP300*) losses among patients below 55 years old, 60–70 years old, and above 70 years old, respectively, whereas five (*KAT2B*, *JAG2*, *HEYL*, *MAML2*, *NOTCH2*), two (*NUMB*, *NCSTN*), and four (*PSENEN*, *SNW1*, *DLL3*, *KAT2A*) genes gained the expression with disease progression, respectively. The comparison of the general trends in expression patterns is presented in Table 2 and Appendix A. Subsequently, we conducted comparative clustering of the Notch targets, and, as shown in the right section of Table 5 and Appendix A, we successfully confirmed many of the primary findings, but there were also some discrepancies.

## 3. Discussion

A well-known and widely used prostate cancer Gleason diagnosis system reflects pathomorphological differentiation of cancer tissue. Such microscopically classified cellular clonality is directly associated with changes in cell molecular activities, meaning differential regulation of gene expression. The Notch signaling pathway is one out of several main regulators of cellular functional differentiation and tissue behavior. In this study, we demonstrated transcriptomic representation of the effects of Notch signaling on PC course and progression according to age of disease onset and tumor stage. We observed that the activity of the core Notch pathway is significantly modulated by both factors, following the natural route of cellular aging, which may be compared to the light dimmer. On one hand, it leads to progressive loss of organic functions and self-renewal capacity in humans [11]. On the other hand, carcinogenic transformation and progression requires uncontrolled proliferation, restrained apoptosis, and the gaining of new functions [12,13]. These seeming dissimilarities are, in fact, two ends of the same stick of common molecular pathways that cause, on one side, loss of organ functionality and, on the other side, confer pro-survival functions at earlier stages of life [14]. In line with the above, we revealed an increase in Notch signaling activity among patients under 55 years of age that intensified with more advanced-stage PC involving the most essential Notch members, such as receptors (*NOTCH2-4*), ligands (*JAG2*, *DLL4*), and *HEYL*, a Notch-specific TF (Figure 1). Among patients aged 60–70 years old, the core Notch pathway was still upregulated, although of a different pattern including *NOTCH1-2* receptors, *JAG1* ligand, *NCSTN* regulator, as well as Notch-related epigenetic transcription modulators such as *HDAC1*, *CREBBP*, and *EP300* (Figure 1). Among the oldest patients, we revealed dimming of the Notch activity through the decrease of expression in its main elements (*JAG1*, *NOTCH1-3*, *CREBBP*, *EP300*, γ-secretase complex) in favor of upregulation of Notch modulators (*RFNG*, *LFNG*, *HDAC2*) (Figure 1). In turn, late-stage PC did not reveal any significant Notch activity, which resembles the Goldilocks Principle, encompassing neither too low, nor too high activity of the pathway to maintain homeostasis [15]. These findings were supported with specific Notch profiles affecting DFS that, importantly, switched with progressing age and stage of the PC patients (Figure 2, Table 3). These observations are, to a large extent, consistent with the current state of knowledge, apart from influence of age and stage, which, as yet, has not been widely considered. For instance, in vitro experiments using PC cell lines revealed that *NOTCH1* activity was promoting cancer cell migration and invasion as well as augmenting aggressiveness of the tumor [16,17]. In humans, a raise in *JAG1* was associated with metastatic PC lesions [18,19], similarly to *NOTCH3*, the expression of which was inversely correlated with survival [20]. Fundamental Notch members were also upregulated in PCs of Gleason 8 compared to the tumors of lower grades [10]. It is worth emphasizing that, among younger patients, we observed an asymmetry between Jagged and Delta ligands, giving cells a rise to a hybrid state described in a series of articles by Boareto et al. In brief, due to Delta–Notch signaling, cells may acquire one of two possible states: Sender (high expression of ligands) or Receiver (high expression of receptors) [21]. However, Delta–Jagged asymmetry enables the cells to acquire a hybrid Sender/Receiver state that implicates a worse clinical outcome. It was shown that during angiogenesis, the asymmetry leads to the hybrid tip/stalk phenotype allowing chaotic and fast sprouting to ensure efficient oxygen supply to rapidly growing tumors [22]. Moreover, it is believed that the hybrid phenotype perpetuates the meta-stable hybrid epithelial/mesenchymal phenotype linked with more aggressive tumor behavior and stem cell-like features [23]. Remarkably, this evidence is a molecular substantiation of the observations of increased PC aggressiveness in younger males and is a proof of being a separate biological entity.

The phenotypic changes of PC induced by Notch are mediated by a transcriptional program of its numerous downstream effectors, including genes linked to invasive properties of cancer that we presented in a previous study [24]. The remarkable changes associating disease progression involve remodeling of cell–cell and cell–ECM interactions such as reshaped adhesion, cell junctions, cytoskeleton components (actin microfilaments, intermediate filaments, microtubules, and molecular motors), protease activity, constituents of the ECM, and differentiation factors determining the fate of the cell and its stem state-related properties [25]. Currently, rapidly growing data confirm the clinical premises of the more aggressive course of early-onset (herein diagnosed at below 55 years of age) PC compared to that in elderly men, which apparently arise from the distinct biology of the tumors [26,27,28,29]. Prior work has implicated age-related specific profiles of EMT, which, at molecular scale, reflected a Gleason grade [30]. In the current research, we performed a much wider description of how age alters tumor invasiveness at cellular resolution; in particular, we emphasized which of these changes are orchestrated by the Notch activity. Each age group showed specific profiles determining the course of PC, as well as a disparate trajectory of evolution associated with stage progression, which, in network analysis, emerged as the four most tightly interconnected clusters (Figure 3). In fact, they could be identified with biological groups because they belong to such as the EMT program, molecular motors, and parts of cell–ECM adhesion (Figure 3).

Several attempts have been made to unveil genomic landscapes of PC in younger and elderly men, although they conferred mostly mutations or signature genes [31,32]. There are well-known risk factors of PC identified as mutations of genes *BRCA1*, *BRCA2*, mismatch repair genes (*MSH2*, *MSH6*, *MLH1*, *PMS2*, *EPCAM*) and *HOXB13* gene coding transcription factor. In our studied cohort of PC patients, the highest mutation frequency has *BRCA2* (5%), *BRCA1* (2.2%), *HOXB13* (1.6%) and *PMS2* (1%), and all others have below 1% mutations. However, these mutation profiles were not related to the molecular and age groups we identified. Conversely, as shown in Figure 5, we deciphered that aggressiveness of early-onset PC originates from reshaping of the cytoskeleton, adhesion, and ECM accompanying EMT-related plasticity of the tumor cells, which was modulated across the progressing stage of PC. The above supports gain in actin microfilaments such as caldesmon (*CALD1*) and myosins (*MYO1D*, *MYO1F*, *MYO7A*) participating in, e.g., assembling the stress fibers and motility-enabling protrusions [33]. Younger patients also showed high activity of microtubule-related molecular motors, i.e., kinesins and dynein, likely arising from uncontrolled cell division, as well as hypoxia-inducible factor 1-α (*HIF1A*). To date, several members of the kinesins family (*KIF11*, *KIF15*, *KIF18B*) have been correlated with poorer prognosis and aggressive forms of PC [34,35,36]. In turn, hypoxia in tumors arises from the metabolic requirements of rapidly dividing cells, which cannot be met due to ineffective neovascularization. However, cancer cells tend to preferentially metabolize glucose over lactate, despite aerobic conditions, known as the Warburg effect. Under hypoxic conditions, HIF1A induces, among others, angiogenesis, the program of EMT, and proliferation, as well as the migration of tumor cells. It is also a key regulator of enzymes involved in aerobic glycolysis [37]. The overexpression of *HIF1A* was, for instance, implicated in the biochemical progression of PC [38], as well as in developing resistance to hormonal therapies in castrate-resistant PC [39].

EMT has been repeatedly demonstrated in the progression and metastasis of many human malignancies [40]. During EMT, epithelial cells gain migratory capacities due to loosening junctions and cell–cell adhesion, as well as remodeling of the cytoskeleton to establish spindle-like morphology. EMT is mainly identified by a loss of major epithelial markers (cytokeratins, E-cadherin) to the detriment of acquiring mesenchymal markers (N-cadherin, vimentin, fibronectin). Importantly, EMT is more and more often considered a hybrid spectrum of epithelial and mesenchymal states, in fact translating to the plasticity that leads to stem cell-like, aggressive cancer phenotypes [41,42,43]. Such cells attaining a hybrid epithelial/mesenchymal phenotype exhibit both epithelial and mesenchymal features (i.e., they are capable of migrating with restricted cell–cell adhesion), which might be a determinant of collective cell migration of multicellular aggregates in the ECM or clustered circulating tumor cells found in the bloodstream of PC patients [44]. Consequently, cells expressing markers of both states have much higher metastatic potential due to enhanced resistance to cell death or more efficient extravasation rates. Thus, this hybrid state resembles the “Mr. Hyde” personality that constitutes much higher metastatic risk in patients than a complete EMT phenotype or “Dr. Jekyll” side [41]. It was further supported by several studies reporting, among others, co-expression of both E-cadherin and vimentin in, e.g., invasive breast cancer, that when exhibited by the tumors concomitantly conferred the worst DFS and OS outcomes [45]. The process of EMT itself may be regulated by many mechanisms and signaling pathways, such as Notch signaling [46], which modulate the expression of EMT-triggering TFs. To date, various members of the EMT program have been reported in PC research in association with phenotypic disparities of the tumor [47]. For instance, Cheaito et al. demonstrated that co-expression of cytokeratin 8 and vimentin correlated with higher Gleason grades and worse prognosis of biochemical recurrence-free survival [48]. In the review devoted to PC plasticity, Papanikolaou and collaborators demonstrated similar findings that were reported on N-cadherin, thus guiding the cross-talk between the tumor stroma and its epithelial cells. EMT has been also associated with the androgen-signaling axis, which is of very high importance in PC, specifically. AR has been considered a transcriptional repressor of E-cadherin in a manner comparable to Snail and Twist TFs, as well as an activator of β-catenin; hence, AR attributed to more mesenchymal phenotypes of PC cells [49].

Figure 5 presents the major findings and how the observed alterations translate into biological consequences for the cell (dark pink elements). In the present study, we noted concurrent overexpression of E-cadherin (*CDH1*) and β-catenin (*CTNNB1*) at the early stage of PC among younger males, which remains in line with our previous studies [30]. It also suggests the phenotype of tumor cells that only partially execute the EMT program, i.e., contemporary expression of both epithelial (*CDH1*, *CTNNA2*, *LAMA1*) and mesenchymal (*FN1*, *CTNNB1*, *MMP7*, *ZEB2*) markers. At the late stage, the cells lost the aforementioned characteristics, thus suggesting a decrease in the invasive potential compared to early-stage tumor cells in men below 55 years of age. In addition, at particular stages of the disease, patients 60–70 and above 70 years old showed very similar features, mostly contrasting to those of patients below 55 years old. Importantly, late-stage tumors of older patients resembled a more aggressive form of PC than in younger patients. Specifically, patients 60–70 years old showed high levels of fibronectin (*FN1*), β-catenin (*CTNNB1*), matrix metalloproteases (*MMP2*, *MMP7*), vimentin (*VIM*), microtubules and molecular motors, and EMT-triggering TFs such as *SNAI2*, *ZEB2*, *TWIST1*, *HIF1A*, and *SMAD3*, whereas the expression of E-cadherin (*CDH1*) was lost. Remarkably, these are the evidence for the predominant mesenchymal phenotype of the tumor cells that executed EMT and acquired migratory potential. Among the oldest patients, we noted heightened E-cadherin (*CDH1*) with concurrent β-catenin (*CTNNB1*), fibronectin (*FN1*), vimentin (*VIM*), microtubules and associated molecular motors, and EMT-related TFs such as *SNAI2*, *ZEB2*, *HIF1A*, and *SMAD3* (detailed results are enclosed in Table 5). Notably, the similar trends observed in the cross-validation study comparing local and regional PCs emphasize the importance of the findings that clearly suggest disparate molecular mechanisms of prostate tumor invasion and progression in association with the age of PC onset. These observations have a number of implications for understanding the clinical behavior of PC diagnosed at different ages and stages, as well as highlight the role of Notch in influencing the plasticity of prostate tumors through modulation of its downstream effectors. These properties are an elicitation of disease progression through executing the sequential program, enabling the cancer cells’ detachment from the primary mass [50].

Much evidence indicates Notch signaling as a potential therapeutical target, and it is currently under clinical trials in some malignancies [51]. However, the reported Notch deregulation and its associations with various aspects of prostate tumorigenesis boils down to specific interactions and single members of the core signaling. Formerly, we confirmed that the carcinogenic effects of Notch are context-specific, i.e., they depend on the intensity of signaling, the pattern of the pathway members expression, and tissue type. The most important finding revealed that the characteristics of each solid tumor, including PC, are represented by specific profiles of the Notch pathway at two levels of signaling: the core and expression of the downstream effectors orchestrated by Notch-specific TFs (HES/HEY families) [52]. In the current study, we correlated the expression of the individual Notch participants with the DFS prognosis among specific age groups, as well as in combination with the PC stage, followed by the evaluation of the downstream effects among Notch targets. Though the current state of knowledge was lacking such findings, of the highest interest was whether the expressions of Notch participants combine into a specific pattern of activation translating into aberrant downstream mechanisms driving aggressive clinical behavior and thus could be identified as a PC-related prognostic factor of unfavorable prognosis. Ultimately, among each group of PC patients, we identified a repeating expression pattern of Notch, translating into significant differences in DFS that were detected in approx. half of the group (Figure 2, Table 3, Appendix A). However, regardless of the known role that each molecule plays during transducing Notch signals, very little is known about their contribution in prostate tumorigenesis and progression. The closest idea to our study was research conducted by Kwon and collaborators that generated a Notch signature score involving the expression of receptors (*NOTCH1-4*), ligands (*JAG1-2*, *DLL1*, *DLL3*, *DLL4*), *DTX1*, and Notch-specific TFs (*HES1-2*, *HEY1-2*). It was demonstrated that the lower the Notch signature, the more favorable the prognosis of disease recurrence (i.e., PSA recurrence-free survival) [53]. In contrast, our signatures included specific mixtures of expression patterns that correlated with better DFS, i.e., *RBPJ*, *RBPJL*, *ADAM17*, *ATNX1*, and *KAT2A*, *LFNG*, *HDAC2*, *SNW1*, *PSEN2* for patients below 55 years old at the early and late stages of PC, respectively; *NOTCH3*, *APH1A*, *HDAC2*, *MFNG*, *DTX4* and *HES5*, *ADAM17*, *CREBBP*, *DVL3*, *ADAM10* for patients 60–70 years old at the early and late stages of PC, respectively; finally, *JAG1*, *PTCRA*, *HEYL*, *DTX3*, *RFNG* for patients above 70 years old at the late stage of the disease (Figure 2, Table 3, Appendix A). Hence, it suggests age-diversified mechanisms of Notch activation drive prostate progressions that reach beyond the fundamental transducers of Notch signaling and shed light on its modulators, affecting the strength of signaling. Following this, we observed that the unfavorable profiles of each signature involved escalated forms of downstream mechanisms of invasion and tumor plasticity. Importantly, they constituted their adversities, along with the disease stage, at the same age of PC onset (Figure 4) and remain consistent with the previous findings of cluster analysis.

## 4. Materials and Methods

For the study, we obtained the RNA-seq expression data with matched clinical information of 502 prostate adenocarcinoma (PC) patients provided by The Cancer Genome Atlas (TCGA). The data were downloaded via the NCBI Gene Expression Omnibus (GEO) series GSE62944 of Rahman et al. that published alternatively reprocessed and compiled RNA-seq with corresponding clinical data for the TCGA samples, with the Rsubread package offering improved analytic performance (normalized TPM values for tumor samples; data status of 27 January 2015) [54]. Then, we restricted the cohort to the specific age groups of ≤55, 60–70, and 70> years old, constituting a total of 397 patients. Additionally, our considerations were focused on the comparison between early-stage tumors restricted to the prostate gland, and late, locally advanced, or metastasizing stage tumors, which were classified with the American Joint Committee on Cancer (AJCC) 8th Edition TNM staging system [55] and subsequently grouped into early (stages I–II) or late (stages III–IV) stage subgroups. The detailed clinical characteristics of the study cohort are shown in Table 7.

The Notch signaling pathway and its core participants in humans were determined according to the KEGG database (hsa04330) [56] and MSigDB v7.2 [57]. The downstream effectors of the Notch signaling comprised a total of 10,740 targets of Notch-specific transcription factors (TFs) belonging to the HES and HEY families (*HES1*, *HES4*, *HES5*, *HEY1*, *HEY2*, *HEYL*) identified through the GTRD database v19.10 [58,59]. The identified target genes haven been annotated with their biological roles and divided into functional groups according to the current knowledge and literature, as well as our previous studies [24,52].

The expression profiling of the Notch core signaling between the groups of patients, according to the combination of age and tumor stage, was performed based on hierarchical clustering of the median gene expression in particular groups with the Pearson distance metric and the complete linkage method. The clustering was performed and visualized with the NMF R package with aheatmap() function. Additionally, to support the hypothesis that Notch signaling differentiates the clinical course of PC among the age groups and in association with the tumor stage, we performed disease-free survival (DFS) analysis regarding the effects of the expression of particular Notch members. The analysis was performed with the EvaluateCutpoints RShiny app through the maxstat algorithm [60]. In addition, we aimed to identify the subgroups of PC patients bearing specific combinations of the Notch core gene profiles associated with DFS prognosis by applying the UpSetR algorithm (UpSetR R package) [61]. For that purpose, we aggregated patients based on intersections of the dummy-encoded expression of particular Notch genes related to DFS outcomes that revealed specific patterns among PC patients. General DFS analysis was subsequently performed to compare the survival of PC patients with and without an identified Notch signature (log-rank test, *p* < 0.05; survminer R package). Furthermore, ExpressCluster software (http://cbdm.hms.harvard.edu/) was used to find common and unique expression profiles of the downstream Notch effectors among the age groups combined with stage of the disease. Clustering was performed by applying the K-means++ algorithm, z-norm (mean = 0, var = 1) signal transformation, rank correlation distance metric, 1000 iterations, and 400 clusters (K), as recommended for six class comparison. Profiles indicating unique contrasts between age groups within the tumor stage, as well as between tumor stages within the age group, were considered as significant. Finally, by using the relevant clusters in terms of functional association with age-related PC course and progression, we constructed the biological networks of interactions with stringApp in Cytoscape [62]. The most pertinent network that we successfully constructed was followed by MCODE identification of highly interconnected regions therein [63]. Moreover, the downstream biological differentiation between the subgroups of patients with specific patterns of Notch signaling was demonstrated by using biological networks from the previous step, with plotted changes in the expression between stage-related groups within the age groups of PC patients.

The findings of the present study have been cross-validated with an independent microarray study deposited in the NCBI GEO (GSE141551) aiming to evaluate the transcriptome-based genetic profiles of 503 localized prostate cancers (PC) associated with the aggressive course concerning age and stage (local vs. regional) of the disease [64]. Due to sample restriction according to the age groups of interest (≤55, 60–70, and 70> years old) we finally employed 380 individuals for validation purpose of the primary findings (local stage, ≤55 years old: 108; regional stage, ≤55 years old: 49; local stage, 60–70 years old: 129; regional stage, 60–70 years old: 69; local stage, 70> years old: 17; regional stage, 70> years old: 8). The validation was performed at two levels: profiling of the Notch core and ExpressCluster analysis of expression profiles of the downstream Notch effectors. Both analyses were performed analogously to the primary ones.

## 5. Conclusions

In a conclusion, PC is a worldwide medical challenge. The heterogeneous course of disease is conditioned by multiple factors, including the age of onset. This factor requires special attention, as aging is considered the most important constituent for cancer development and, importantly, determines decisions on the treatment strategy applied. While distinguishing the aggressive from the indolent form of PC is problematic, we characterized the molecular determinants of the age-related clinical behavior of the tumors. Primarily, it stems from different invasive properties depending on the route of the EMT program and adherent remodeling of cellular architecture. The Notch pathway, one of the most conservative mechanisms of signaling, through specific patterns of activation, orchestrates transcriptional programs corresponding to PC outcomes. While excessive Notch signaling in younger patients predestines more aggressive forms of PC, insufficiency in the signaling worsens the prognosis in elderly men. Of utmost relevance is the discovery of age- and stage-specific combinations of the Notch molecules, herein described as Notch signatures, which predict unfavorable outcomes arising from escalated invasive features, indicating an undifferentiated, stem cell-like state of the tumor cells and could constitute a new prognostic and therapeutic approach for prostate cancers.

## Figures and Tables

**Figure 1 ijms-24-00164-f001:**
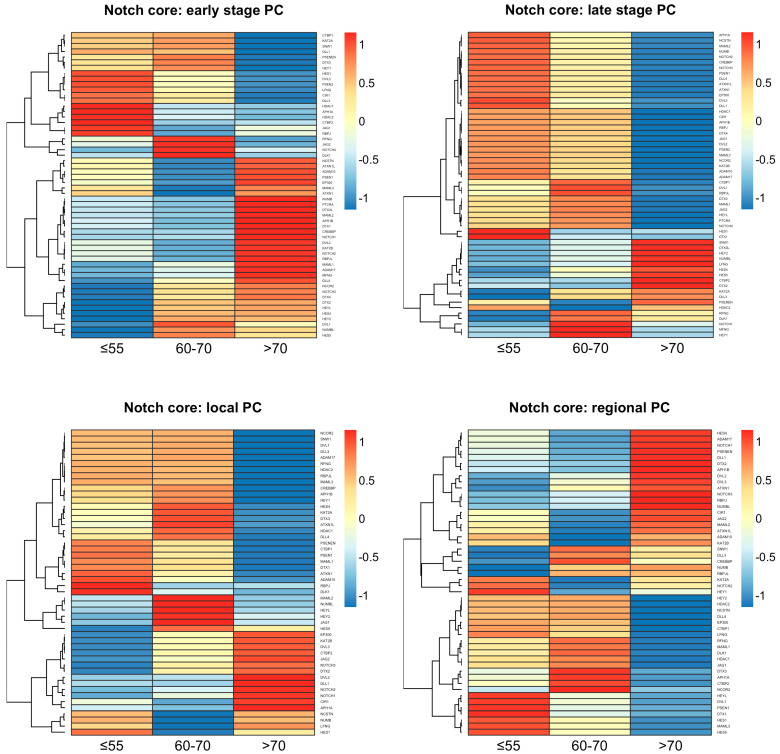
The profiles of core Notch signaling differentiate age subgroups of PC.

**Figure 2 ijms-24-00164-f002:**
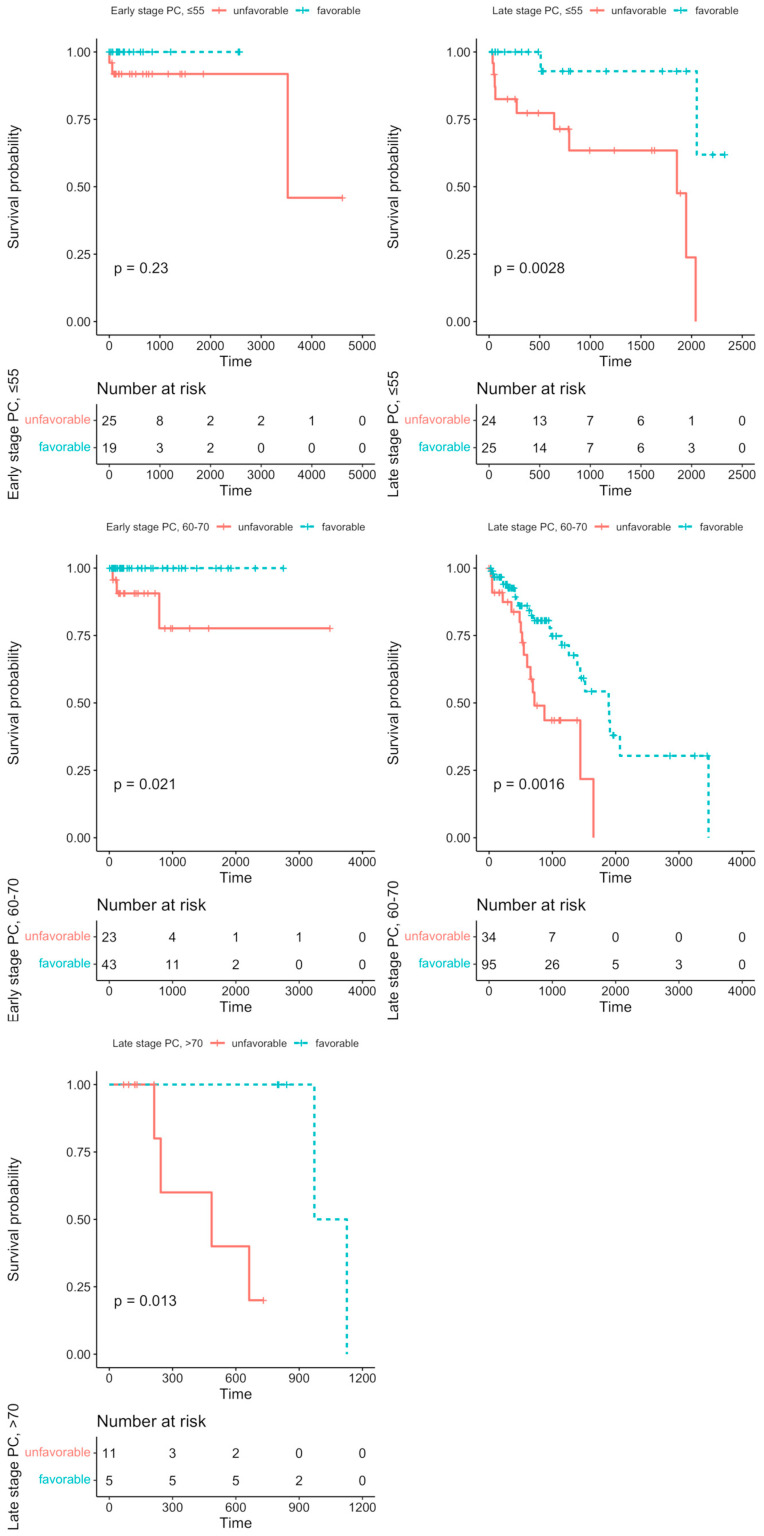
The specified Notch patterns of expression altered the DFS among subgroups of PC patients through the modulation of reshaping the cellular architecture. The unfavorable prognosis of signatures consisting of *RBPJ*, *ADAM17*, *RBPJL*, *ATXN1* in early-stage PC below 55 years old, *KAT2A*, *LFNG*, *HDAC2*, *SNW1*, *PSEN2* in late-stage PC below 55 years old, *NOTCH3*, *APH1A*, *HDAC2*, *MFNG*, *DTX4* in early-stage PC among 60–70-year-olds, *HES5*, *ADAM17*, *CREBBP*, *DVL3*, *ADAM10* in late-stage PC among 60–70-year-olds, and *JAG1*, *PTCRA*, *HEYL*, *DTX3*, *RFNG* in late-stage PC above 70 years old. Firstly, the DFS analysis was performed regarding the effects of the expression of particular Notch members through the maxstat algorithm, determining the optimal expression cutpoint stratifying the survival in the most significant manner. Next, by applying the UpSetR algorithm, we identified the specific subgroups of PC patients bearing a particular combination of the Notch core gene profiles associated with the DFS prognosis.

**Figure 3 ijms-24-00164-f003:**
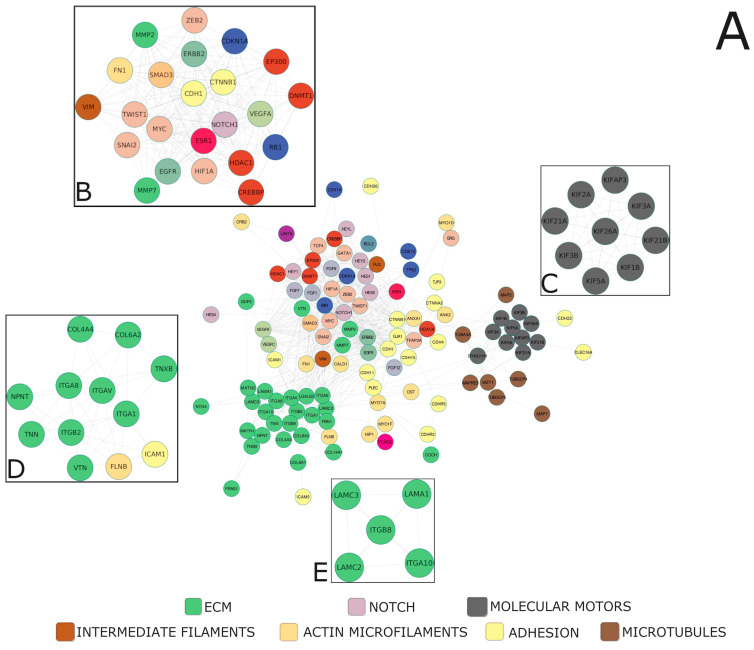
The functional network of the major molecules underlying invasive properties of PC with highly interconnected compartments (**A**). MCODE clustering indicated four of the most significant biological subgroups underlying age-associated course of PC, i.e., (**B**) the key participants of EMT program, (**C**) microtubule-related molecular motors, (**D**,**E**) parts of cell-ECM adhesion and interactions. We applied the clustering (ExpressCluster) to find common and unique expression profiles of the downstream Notch effectors among the age groups combined with the stage of the disease. Furthermore, by using the relevant clusters in terms of functional association with age-related PC course and progression, we constructed the biological networks of interactions, followed by MCODE identification of highly interconnected regions therein.

**Figure 4 ijms-24-00164-f004:**
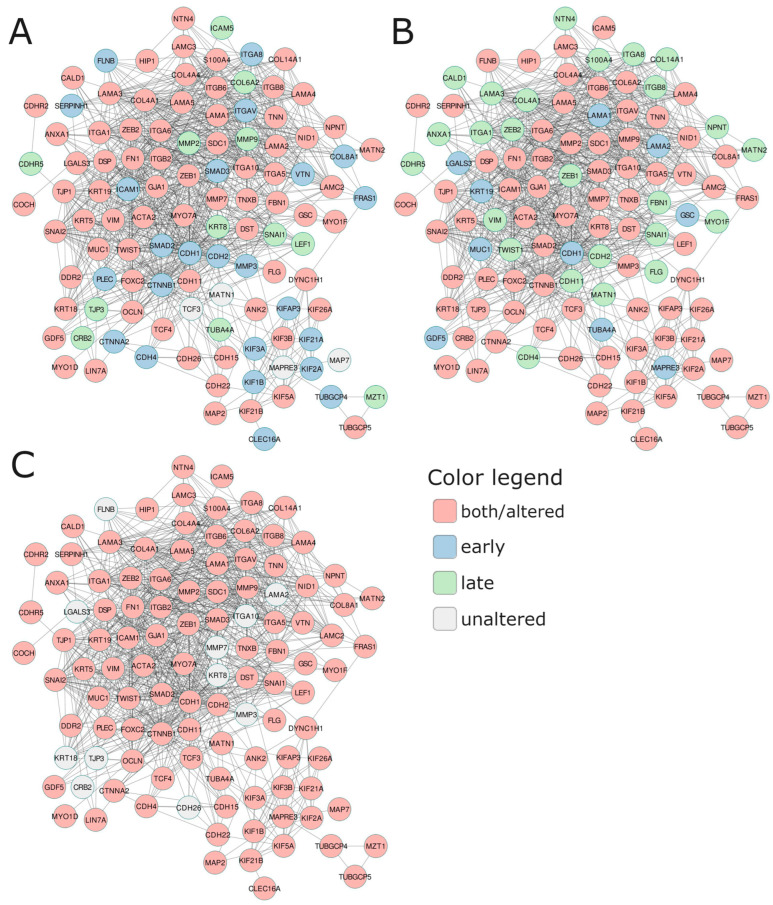
The functional network comprising the most essential invasion-related molecules shows expression profiles shared by (**A**) early-onset PC, (**B**) 60–70-year-old PC patients, and (**C**) elderly PC patients, reflecting the DFS prognosis according to the specific Notch signature. The adjacent Table 6 presents fold change (FC) of gene expression across different subgroups of patients.

**Figure 5 ijms-24-00164-f005:**
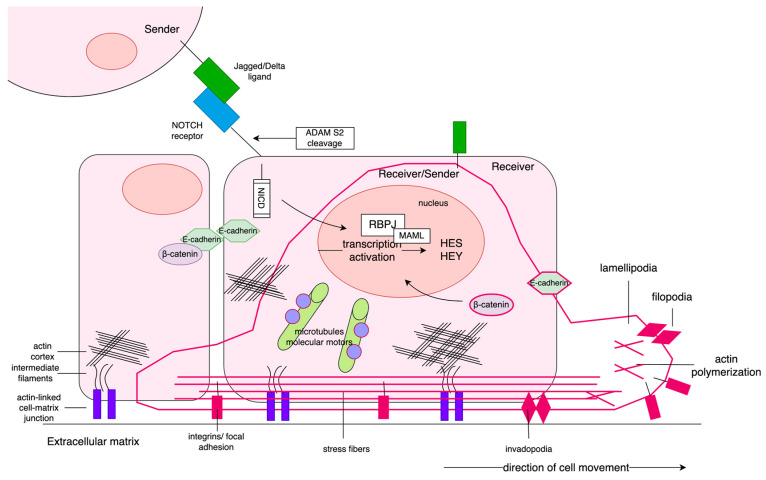
PC occurring in males below 55 years old showed intense activation of Notch signaling of Sender phenotype at early stage of disease, and, especially, at the late stage of disease through acquisition of hybrid Sender/Receiver phenotype (high expression of both Notch ligands and receptors). In turn, among males over 70 years old, Notch signaling has been diminished and turned into Receiver phenotype of PC cells at early stage of disease, whereas at the late-stage Notch has been inactivated. Downstream effects governed by Notch-specific TFs (HES/HEY) involved a change of cell morphology through reorganization of cellular architecture. The sedentary epithelial cells have apical/basal polarity, the cytoskeleton comprises the actin cortex, cytokeratin intermediate filaments, and microtubules with dynein and kinesin motors. The cell is anchored to the ECM by actin-linked laminin junctions, and the between-cells adhesion is asserted by E-cadherin bridges. Due to Notch signaling in younger males, PC cells undergo partial EMT triggered by *ZEB2* concomitantly expressing epithelial (*CDH1*) and mesenchymal (*CTNNB1*, *FN1*) markers. As demonstrated by the elements in dark pink, consequently, the cells start to migrate, pushed forward by contractions via stress fibers and detachment from the matrix eased by the activity of *MMP7*. Finally, at the leading edge, the invading cells adhere by focal adhesion (involving *LAMA1*). Many of these features have been lost with disease progression, and tough gain in VIM has been noticed. The PC cells in older males demonstrated, in turn, inverse profiles to those of patients below 55 years old. Moreover, we observed a much-intensified activation of the EMT program through involvement of all specific TFs (*SNAI2*, *TWIST1*, *ZEB2*, *HIF1A*, *SMAD3*) at both stages of PC. Late-stage tumors restored the expression of *CDH1,* simultaneously maintaining *CTNNB1* and *VIM* independent of PC stage. Tumors of both stages kept adhesive molecules like integrins and collagens. Importantly, high activity of microtubules and kinesin/dynein was kept.

**Table 1 ijms-24-00164-t001:** A detailed comparison of the Notch expression between tumor stages according to the age of PC onset.

		Age Group					
		≦55		60–70		>70	
		Early	Late	Early	Late	Early	Late
UP	core	ligands: *DLL1*, *DLL3*, *JAG1*, TFs: *HEY1*, *HES1*, *RBPJ*, modulators: *PSENEN*, *PSEN2*, *LFNG*, *APH1A*, *NCSTN*, *ADAM10*, *PSEN1*, *MAML3*	ligands: *DLL1*, *DLL4*, *JAG1*, *JAG2*, receptors: *NOTCH2*, *NOTCH3*, *NOTCH4*, TFs: *RBPJ*, *RBPJL*, *HEYL*, *PTCRA*, *HES1*, modulators: *APH1A*, *NCSTN*, *MAML2*, *NUMB*, *PSEN1*, *APH1B*, *PSEN2*, *MAML3*, *ADAM10*, *ADAM17*, *MAML1*, *PSENEN*	ligands: *DLL1*, *DLL3*, *DLL4*, *JAG2*, *DLK1*, receptors: *NOTCH3*, *NOTCH4*, TFs: *HEY1*, *HES1*, *HEYL*, *HES4*, *HEY2*, *HES5* modulators: *PSENEN*, *PSEN2*, *LFNG*, *RFNG*, *NUMBL*	ligands: *DLL4*, *DLL1*, *JAG1*, *JAG2*, *DLL3*, *DLK1*, receptors: *NOTCH1*, *NOTCH2*, *NOTCH3*, *NOTCH4*, TFs: *RBPJ*, *RBPJL*, *HEYL*, *PTCRA*, *HES4*, *HEY1*, modulators: *APH1A*, *NCSTN*, *MAML2*, *NUMB*, *PSEN1*, *APH1B*, *PSEN2*, *MAML3*, *ADAM10*, *ADAM17*, *MAML1*, *RFNG*, *MFNG*	ligands: *JAG1*, *DLL4*, receptors: *NOTCH1*, *NOTCH2*, *NOTCH3*, TFs: *RBPJ*, *PTCRA*, *RBPJL*, *HEYL*, *HES4*, *HEY2*, *HES5*, modulators: *NCSTN*, *ADAM10*, *PSEN1*, *MAML3*, *NUMB*, *MAML2*, *APH1B*, *MAML1*, *ADAM17*, *MFNG*, *NUMBL*	ligands: *DLK1*, *DLL3*, receptors: TFs: *HEY2*, *HES4*, *HES5*, modulators: *NUMBL*, *LFNG*, *PSENEN*, *RFNG*
	cross-talk	epigenetic modulators: *HDAC1*, *HDAC2*, *EP300*, regulators: *CTBP1*, *CTBP2*, *KAT2A*, *SNW1*, *DTX3*, *CIR1*, *ATXN1L*, Wnt cross-talk: *DVL3*	epigenetic modulators: *CREBBP*, *EP300*, *HDAC1*, *HDAC2*, regulators: *ATXN1L*, *ATXN1*, *CIR1*, *DTX1*, *DTX3*, *DTX4*, *KAT2B*, *NCOR2*, *CTBP1*, Wnt cross-talk: *DVL1*, *DVL2*, *DVL3*	regulators: *CTBP1*, *KAT2A*, *SNW1*, *DTX3*, *CIR1*, *NCOR2*, *DTX4*, *DTX2*, Wnt cross-talk: *DVL1*, *DVL3*	epigenetic modulators: *CREBBP*, *EP300*, *HDAC1*, regulators: *ATXN1L*, *ATXN1*, *CIR1*, *DTX4*, *NCOR2*, *KAT2B*, *CTBP1*, *DTX3*, *KAT2A*, Wnt cross-talk: *DVL3*, *DVL2*, *DVL1*	epigenetic modulators: *EP300*, *CREBBP*, regulators: *ATXN1L*, *ATXN1L*, *DTX3L*, *DTX1*, *KAT2B*, *NCOR2*, *DTX4*, *DTX2*, Wnt cross-talk: *DVL2*, *DVL1*	epigenetic modulators: *HDAC2*, regulators: *SNW1*, *DTX3L*, *CTBP2*, *DTX2*, *KAT2A*
DOWN	core	ligands: *JAG2*, *DLL4*, *DLK1* receptors: *NOTCH4*, *NOTCH1*, *NOTCH2*, *NOTCH3*, TFs: *PTCRA*, *RBPJL*, *HEYL*, *HES4*, *HEY2*, *HES5*, modulators: *RFNG*, *NUMB*, *MAML2*, *APH1B*, *MAML1*, *ADAM17*, *MFNG*, *NUMBL*	ligands: *DLK1*, *DLL3*, receptors: *NOTCH1* TFs: *HEY2*, *HES4*, *HES5*, *HEY1*, modulators: *NUMBL*, *LFNG*, *RFNG*, *MFNG*	ligands: *JAG1*, receptors: *NOTCH1*, *NOTCH2*, TFs: *RBPJ*, *PTCRA*, *RBPJL*, modulators: *APH1A*, *NCSTN*, *ADAM10*, *PSEN1*, *MAML3*, *NUMB*, *MAML2*, *APH1B*, *MAML1*, *ADAM17*, *MFNG*	TFs: *HES1*, *HEY2*, *HES5*, modulators: *NUMBL*, *LFNG*, *PSENEN*	ligands: *DLL1*, *DLL3*, *JAG2*, *DLK1*, receptors: *NOTCH4*, TFs: *HEY1*, *HES1*, modulators: *PSENEN*, *PSEN2*, *LFNG*, *APH1A*, *RFNG*	ligands: *DLL4*, *DLL1*, *JAG1*, *JAG2*, receptors: *NOTCH2*, *NOTCH3*, *NOTCH4*, *NOTCH1*, TFs: *RBPJ*, *RBPJL*, *HEYL*, *PTCRA*, *HES1*, *HEY1*, modulators: *APH1A*, *NCSTN*, *MAML2*, *NUMB*, *PSEN1*, *APH1B*, *PSEN2*, *MAML3*, *ADAM10*, *ADAM17*, *MAML1*, *MFNG*
	cross-talk	epigenetic modulators: *CREBBP*, regulators: *DTX3L*, *DTX1*, *KAT2B*, *NCOR2*, *DTX4*, *DTX2*, Wnt cross-talk: *DVL2*, *DVL1*,	regulators: *SNW1*, *DTX3L*, *CTBP2*, *DTX2*, *KAT2A*	epigenetic modulators: *HDAC1*, *HDAC2*, *EP300*, *CREBBP*, regulators: *CTBP2*, *ATXN1L*, *ATXN1L*, *DTX3L*, *DTX1*, *KAT2B*, Wnt cross-talk: *DVL2*	epigenetic modulators: *HDAC2*, regulators: *DTX1*, *SNW1*, *DTX3L*, *CTBP2*, *DTX2*	epigenetic modulators: *HDAC1*, *HDAC2*, regulators: *CTBP1*, *KAT2A*, *SNW1*, *DTX3*, *CIR1*, *CTBP2*, Wnt cross-talk: *DVL3*	epigenetic modulators: *CREBBP*, *EP300*, *HDAC1*, regulators: *ATXN1L*, *ATXN1*, *CIR1*, *DTX4*, *NCOR2*, *KAT2B*, *CTBP1*, *DTX3*, *DTX1*, Wnt cross-talk: *DVL3*, *DVL2*, *DVL1*

**Table 2 ijms-24-00164-t002:** Corroboration of the primary findings with validation cohort (detailed lists of genes are available in Appendix A).

Disease Stage	Expression Pattern	In	Age Group		
			≤55	60–70	70>
**General Trends**					
early/local	UP	TCGA only	7	8	25
		Rubicz et al. only	14	19	8
		Overlapping trend	18	20	10
	DOWN	TCGA only	18	23	10
		Rubicz et al. only	6	5	21
		Overlapping trend	12	6	11
late/regional	UP	TCGA only	19	26	8
		Rubicz et al. only	8	7	20
		Overlapping trend	21	18	7
	DOWN	TCGA only	10	8	25
		Rubicz et al. only	15	21	7
		Overlapping trend	6	4	16
**Shift in Expression between Stages**					
early -> late local -> regional	UP -> DOWN	TCGA only	6	8	28
		Rubicz et al. only	13	18	7
		Overlapping trend	2	3	2
	DOWN -> UP	TCGA only	20	23	9
		Rubicz et al. only	10	5	16
		Overlapping trend	5	2	4

**Table 3 ijms-24-00164-t003:** The core participants of the Notch pathway differentiate DFS according to patients’ age and stage of the disease. The number represents the hazard ratio (HR) and the color indicates the trend–the blue color indicates that lower expression (i.e., below the determined cutpoint) was more favorable, and the red indicates that higher expression (i.e., above the determined cutpoint) was more favorable regarding DFS prognosis. Legend: ^1^ All patients were tumor-free; * *p* < 0.05, ** *p* < 0.01, *** *p* < 0.001.

	≤55			60–70			70>		
	General HR	Early HR	Late HR	General HR	Early HR	Late HR	General HR	Early HR ^1^	Late HR
*ADAM10*			5.22 *		14.4 **	0.373 *			
*ADAM17*	3.16 *	100> *	4.46 *			0.455 *			
*APH1A*				2.75 ***	15.1 **	1.9 *			
*APH1B*	0.29 *						0.069 *		
*ATXN1*		100> *					<0.001 **		<0.001 *
*ATXN1L*				0.14 *					
*CIR1*	3.01 *						0.112 *		
*CREBBP*						0.402 *			
*CTBP1*				2.97 *					
*CTBP2*				0.251 *		<0.001 **			
*DLK1*				2.15 *		2.26 *			
*DLL1*				0.35 *		0.32 **			
*DLL3*				0.482 *	0.12 *				
*DLL4*							0.069 *		
*DTX1*	5.64 *						<0.001 *		
*DTX2*				0.347 *	<0.001 **				
*DTX3*				1.91 *			0.0874 *		0.107 *
*DTX3L*				2.85 **	11.2 *	0.515 *			
*DTX4*					0.0873 *				
*DVL1*									
*DVL2*									
*DVL3*	4.91 *			2.18 **	9.73 *	0.299 **	9.54 *		
*EP300*						<0.001 *			
*HDAC1*					9.4 *	0.398 *			
*HDAC2*			3.19 *		12.7 **				
*HES1*						0.451 *			
*HES4*									
*HES5*						0.339 **	100> **		
*HEY1*	0.296 *						0.157 *		
*HEY2*									
*HEYL*			0.284 *	1.86 *					100> **
*JAG1*							0.079 **		0.093 *
*JAG2*	3.03 *			6.12 *		1.94 *	0.069 *		
*KAT2A*		100> *	3.58 *		100> *				
*KAT2B*						0.382 **	0.107 *		
*LFNG*			6.77 **						<0.001 ***
*MAML1*									
*MAML2*									
*MAML3*							<0.001 *		
*MFNG*					0.0539 ***				
*NCOR2*									
*NCSTN*						0.434 **	0.0954 **		
*NOTCH1*									
*NOTCH2*				1.89 *					
*NOTCH3*				2.37 **	24.9 ***				
*NOTCH4*	3.34 *						0.069 *		
*NUMB*	3.25 *		5.65 *						
*NUMBL*									
*PSEN1*	3.48 *		3.87 *			0.44 **			
*PSEN2*	0.234 *		0.226 *	1.91 *	<0.001 *		<0.001 *		<0.001 **
*PSENEN*	6.76 *			0.099 **		0.251 *			
*PTCRA*					<0.001 *				12.3 **
*RBPJ*		100> *				<0.001 *			
*RBPJL*		100> *							
*RFNG*							0.0874 *		0.107 *
*SNW1*	4.21 *		7.62 ***	1.95 *	14.1 **				

**Table 4 ijms-24-00164-t004:** Cluster analysis revealed molecular alterations affecting invasive properties of PC driven by the Notch pathway. Blue color and “↓” indicates downregulation of the expression, red color and “↑” indicates upregulation of the expression, and NA indicates not available expression profile of a particular gene. All the resulting clusters are available in the Appendix A.

		RNA-Seq						Microarrays					
		Early			Late			Local			Regional		
		≤55	60–70	70>	≤55	60–70	70>	≤55	60–70	70>	≤55	60–70	70>
*ANK2*	actin microfilaments	↓	↑	↑	↓	↑	↑	↓	↑	↓	↓	↑	↓
*ANXA1*		↓	↑	↑	↓	↓	↑	↓	↑	↓	↓	↑	↓
*CALD1*		↑	↓	↑	↓	↓	↑	↑	↑	↑	↓	↓	↓
*CRB2*		↑	↓	↑	↑	↓	↑	↑	↑	↓	↑	↑	↓
*DST*		↑	↓	↓	↓	↑	↑	↑	↑	↓	↓	↓	↑
*FLNB*		↑	↓	↓	↓	↑	↑	↑	↑	↓	↑	↑	↓
*FN1*		↑	↓	↓	↓	↑	↑	↓	↑	↓	↓	↑	↓
*HIP1*		↑	↑	↓	↑	↓	↓	↓	↓	↓	↑	↑	↑
*MYO1D*		↑	↓	↓	↓	↑	↑	↑	↑	↓	↑	↑	↓
*MYO1F*		↑	↑	↓	↑	↑	↑	↓	↓	↑	↓	↓	↑
*MYO7A*		↓	↓	↓	↓	↓	↑	↑	↓	↓	↑	↓	↑
*CDH1*	adhesion	↑	↓	↑	↓	↓	↑	↑	↓	↓	↑	↓	↓
*CDH11*		↓	↓	↓	↓	↑	↑	↓	↓	↑	↓	↓	↑
*CDH15*		↓	↓	↑	↑	↑	↓	↑	↑	↓	↑	↓	↓
*CDH22*		↑	↑	↓	↓	↓	↓	NA					
*CDH26*		↓	↑	↑	↓	↑	↑	↑	↑	↑	↑	↑	↓
*CDH4*		↓	↓	↑	↑	↑	↓	↓	↓	↓	↑	↑	↓
*CDHR2*		↑	↓	↑	↑	↓	↓	↓	↑	↓	↓	↓	↑
*CDHR5*		↑	↓	↓	↑	↓	↓	↑	↑	↓	↑	↑	↓
*CLEC16A*		↑	↓	↓	↓	↑	↑	↓	↑	↑	↑	↓	↓
*CTNNA2*		↓	↑	↑	↑	↑	↓	↑	↑	↑	↓	↑	↓
*CTNNB1*		↑	↓	↓	↓	↑	↑	↑	↑	↓	↓	↑	↑
*GJA1*		↓	↑	↓	↓	↑	↑	↓	↓	↑	↓	↓	↑
*ICAM1*		↑	↓	↓	↓	↓	↑	↓	↑	↓	↓	↑	↑
*ICAM5*		↑	↑	↓	↑	↑	↓	↓	↑	↓	↓	↑	↑
*PLEC*		↑	↓	↓	↓	↑	↑	↑	↑	↓	↑	↓	↓
*TJP3*		↑	↓	↓	↓	↓	↑	↓	↓	↑	↓	↓	↑
*VEGFA*	angiogenesis	↑	↓	↑	↓	↓	↑	↓	↓	↑	↓	↓	↑
*VEGFC*		↑	↓	↓	↑	↓	↓	↓	↑	↓	↑	↑	↑
*BCL2*	apoptosis	↑	↓	↓	↓	↑	↑	↓	↓	↑	↓	↓	↑
*CDK12*	cell cycle	↑	↓	↓	↓	↓	↑	↑	↑	↓	↑	↑	↓
*CDK19*		↓	↓	↑	↓	↑	↑	↓	↓	↑	↑	↑	↓
*CDKN1A*		↑	↑	↓	↑	↓	↓	↓	↑	↓	↓	↑	↓
*RB1*		↑	↓	↓	↓	↑	↑	↓	↓	↑	↓	↓	↑
*TP63*		↓	↑	↑	↓	↑	↑	↓	↓	↑	↓	↓	↑
*LIN7A*	cell polarity	↓	↑	↓	↑	↑	↓	↓	↓	↓	↓	↑	↑
*COCH*	ECM	↓	↓	↑	↓	↓	↑	↓	↓	↑	↓	↓	↑
*COL14A1*		↑	↓	↓	↓	↑	↑	↓	↑	↓	↓	↑	↓
*COL4A4*		↑	↓	↓	↓	↓	↑	↑	↑	↑	↓	↓	↓
*COL6A2*		↑	↑	↓	↓	↑	↑	↑	↑	↓	↓	↓	↓
*COL8A1*		↑	↓	↓	↓	↑	↑	↓	↓	↑	↓	↓	↑
*FBN1*		↑	↓	↓	↓	↑	↑	↑	↑	↓	↑	↓	↑
*FRAS1*		↓	↑	↓	↓	↑	↑	↓	↑	↓	↑	↑	↑
*GDF5*		↑	↓	↑	↑	↓	↑	↑	↓	↑	↓	↑	↑
*ITGA1*		↑	↓	↓	↓	↑	↑	↓	↓	↑	↓	↓	↑
*ITGA10*		↓	↑	↑	↓	↑	↑	↓	↓	↑	↓	↓	↑
*ITGA6*		↑	↓	↑	↓	↑	↑	↓	↓	↑	↓	↓	↑
*ITGA8*		↑	↓	↓	↓	↑	↑	↓	↓	↑	↓	↓	↑
*ITGAV*		↑	↓	↓	↓	↑	↑	↓	↓	↑	↑	↓	↑
*ITGB2*		↑	↓	↓	↓	↑	↑	↓	↓	↑	↓	↓	↑
*ITGB8*		↓	↑	↑	↓	↓	↑	↓	↓	↑	↑	↓	↓
*LAMA1*		↑	↓	↓	↑	↓	↓	↓	↓	↑	↑	↓	↑
*LAMC2*		↓	↑	↑	↓	↓	↑	↑	↑	↓	↑	↑	↓
*LAMC3*		↓	↑	↓	↓	↑	↑	↓	↓	↑	↓	↓	↑
*LGALS3*		↑	↓	↓	↓	↓	↓	↑	↑	↑	↑	↓	↓
*MATN1*		↑	↓	↑	↑	↓	↑	↓	↓	↓	↓	↑	↑
*MATN2*		↓	↑	↑	↓	↑	↑	↓	↓	↑	↑	↓	↑
*MMP7*		↑	↓	↓	↓	↑	↑	↓	↑	↑	↓	↓	↓
*NPNT*		↑	↓	↓	↑	↓	↓	↑	↑	↓	↓	↑	↓
*NTN4*		↓	↑	↑	↓	↑	↑	↓	↓	↑	↑	↓	↓
*TNN*		↓	↓	↓	↑	↓	↑	↑	↑	↓	↑	↑	↓
*TNXB*		↑	↓	↓	↓	↑	↑	NA					
*VTN*		↓	↑	↓	↑	↑	↓	NA					
*MMP2*		↓	↑	↓	↓	↑	↑	↓	↓	↓	↓	↓	↑
*EGFR*	EGFR signaling	↑	↓	↓	↓	↑	↑	↓	↓	↑	↓	↓	↑
*ERBB2*		↓	↑	↑	↓	↑	↑	↑	↑	↓	↑	↑	↓
*FCHO2*	endocytosis	↑	↓	↑	↓	↑	↑	↓	↓	↑	↓	↓	↑
*ESR1*	estrogen	↑	↑	↓	↓	↑	↑	↑	↑	↓	↓	↓	↓
*FGF1*	FGF signaling	↑	↑	↓	↓	↑	↑	↓	↑	↓	↓	↓	↑
*FGF12*		↓	↑	↑	↓	↑	↑	↓	↓	↑	↑	↑	↓
*FGF7*		↑	↓	↓	↑	↓	↓	NA					
*FGF9*		↑	↓	↓	↑	↓	↑	↓	↑	↑	↓	↑	↓
*CREBBP*	epigenetic regulators	↑	↓	↓	↓	↑	↑	↑	↑	↓	↓	↑	↓
*DNMT1*		↑	↓	↓	↑	↑	↑	↓	↓	↑	↓	↓	↑
*EP300*		↑	↓	↓	↓	↑	↑	↓	↓	↑	↑	↓	↑
*HDAC1*		↓	↓	↑	↓	↑	↑	↑	↑	↓	↑	↑	↓
*HDAC6*		↑	↓	↑	↓	↑	↑	↑	↑	↓	↓	↓	↓
*FLG*	intermediate filaments	↑	↓	↑	↓	↓	↑	↑	↑	↓	↑	↑	↓
*VIM*		↓	↑	↓	↓	↑	↑	↑	↑	↓	↓	↓	↑
*MAP2*	microtubules	↑	↓	↓	↓	↓	↓	↓	↓	↑	↓	↓	↑
*MAP7*		↓	↓	↑	↓	↑	↑	↓	↓	↑	↑	↓	↑
*MAPRE3*		↓	↓	↑	↓	↑	↑	↓	↑	↓	↓	↓	↑
*MZT1*		↓	↓	↑	↓	↑	↑	↓	↓	↑	↑	↑	↓
*TUBA4A*		↓	↑	↑	↓	↑	↑	↓	↓	↑	↓	↓	↑
*TUBGCP4*		↑	↓	↑	↓	↑	↑	NA					
*TUBGCP5*		↑	↓	↑	↓	↓	↑	NA					
*DYNC1H1*	molecular motors	↑	↓	↓	↓	↑	↑	↓	↑	↓	↓	↑	↓
*KIF1B*		↑	↓	↓	↓	↑	↑	↑	↑	↓	↑	↓	↑
*KIF21A*		↑	↓	↓	↓	↑	↑	↑	↑	↓	↑	↑	↓
*KIF21B*		↑	↓	↓	↓	↑	↑	↓	↓	↑	↓	↓	↑
*KIF26A*		↑	↑	↓	↓	↑	↑	NA					
*KIF2A*		↑	↓	↓	↓	↑	↑	NA					
*KIF3A*		↑	↓	↓	↓	↑	↑	↓	↑	↓	↓	↑	↓
*KIF3B*		↑	↓	↓	↓	↓	↓	↓	↑	↓	↑	↑	↑
*KIF5A*		↓	↓	↓	↓	↓	↑	↓	↓	↓	↓	↑	↑
*KIFAP3*		↑	↓	↓	↓	↑	↑	↓	↓	↑	↓	↓	↑
*HES1*	Notch	↓	↓	↑	↑	↑	↑	↓	↓	↑	↑	↓	↓
*HES4*		↓	↑	↓	↑	↑	↓	↓	↑	↓	↓	↑	↓
*HES5*		↓	↑	↑	↑	↑	↑	↓	↑	↑	↑	↓	↓
*HEY1*		↓	↑	↓	↑	↑	↓	↓	↑	↓	↓	↑	↑
*HEY2*		↓	↓	↓	↑	↑	↓	↑	↑	↓	↑	↑	↓
*HEYL*		↓	↑	↓	↑	↑	↑	↓	↑	↓	↑	↓	↑
*NOTCH1*		↑	↑	↓	↓	↑	↑	↓	↓	↑	↓	↓	↑
*ERG*	TF	↓	↓	↑	↓	↑	↑	↑	↑	↓	↑	↑	↓
*GATA1*		↑	↓	↓	↓	↓	↑	↑	↑	↓	↑	↑	↓
*MYC*		↑	↓	↓	↑	↑	↓	↓	↑	↑	↓	↑	↓
*TCF4*		↑	↓	↓	↓	↑	↑	↓	↓	↑	↓	↓	↑
*TFAP2A*		↓	↑	↑	↓	↑	↑	↑	↓	↓	↑	↑	↓
*SNAI2*	TF/EMT	↓	↑	↑	↓	↑	↑	↓	↓	↑	↓	↓	↑
*TWIST1*		↓	↑	↓	↑	↑	↓	↑	↓	↓	↑	↑	↑
*ZEB2*		↑	↓	↓	↓	↑	↑	↓	↓	↑	↓	↓	↑
*HIF1A*	TF/hypoxia	↑	↓	↓	↓	↑	↑	↓	↑	↓	↓	↑	↓
*SMAD3*	TGFβ signaling/EMT	↓	↑	↓	↓	↑	↑	↑	↑	↑	↓	↓	↓

**Table 5 ijms-24-00164-t005:** The Notch signature specific for each PC subgroup significantly differentiates DFS.

		Notch Signature	Prognosis	n		rmean * Survival	SE(rmean) *	Median Survival (95% CI *)	HR* (95% CI)	*p*
				Patients	Events					
≦55 years old	early stage	*RBPJ*, *RBPJL*, *ADAM17*, *ATXN1*	unfavorable	25	3	3734.3	415.8	3524 (3524-NA)	2.23 × 10^−09^ (0-Inf)	0.23
			favorable	19	0	2576	0	NA	0.08 (0.01–0.64)	
	late stage	*KAT2A*, *LFNG*, *HDAC2*, *SNW1*, *PSEN2*	unfavorable	24	10	1366.8	185.4	1855 (791-NA)	8 × 10^−10^ (0-Inf)	0.0028
			favorable	25	2	2122.4	137.9	NA (2051-NA)	0.35 (0.18–0.69)	
60–70 years old	early stage	*NOTCH3*, *APH1A*, *HDAC2*, *MFNG*, *DTX4*	unfavorable	23	3	2819	375.4	NA	3.5 × 10^−10^ (0-Inf)	0.021
			favorable	43	0	2753	0	NA	2.23 × 10^−09^ (0-Inf)	
	late stage	*HES5*, *ADAM17*, *CREBBP*, *DVL3*, *ADAM10*	unfavorable	34	16	952.9	115.2	717 (602-NA)	0.08 (0.01–0.64)	0.0016
			favorable	95	25	1933	210	1890 (1395-NA)	8 × 10^−10^ (0-Inf)	
>70 years old	late stage	*JAG1*, *PTCRA*, *HEYL*, *DTX3*, *RFNG*	unfavorable	11	4	467.2	94.4	485 (244-NA)	0.35 (0.18–0.69)	0.013
			favorable	5	2	1049	54.4	1049 (972-NA)	3.5 × 10^−10^ (0-Inf)	

* rmean survival—restricted mean survival time; SE(rmean)—standard error of restricted mean survival time; 95% CI—95% confidence intervals; HR—hazard ratio.

**Table 6 ijms-24-00164-t006:** The complementary data underlying networks in Figure 4 presenting expression profiles of the most essential invasion-related molecules that reflect the DFS prognosis according to the specific Notch signature. The values represent fold changes (FC) between unfavorable vs. favorable prognosis according to the specified Notch patterns.

		≦55		60–70		>70
		Early	Late	Early	Late	Late
actin microfilament	*ACTA2*	0.84	0.38	0.48	0.88	0.32
	*ANK2*	1.17	0.56	0.86	0.49	0.38
	*ANXA1*	0.76	0.74	0.91	0.89	0.61
	*CALD1*	1.32	0.76	1.07	0.54	0.34
	*CRB2*	1.07	1.25	1.23	0.77	0.92
	*DST*	1.82	0.81	0.80	0.60	0.87
	*FLNB*	1.57	1.06	1.39	0.90	1.03
	*FN1*	1.42	0.72	1.11	0.47	0.50
	*HIP1*	1.28	1.14	1.18	0.69	0.69
	*MYO1D*	1.28	0.86	1.25	0.85	0.76
	*MYO1F*	1.73	0.78	0.94	0.65	1.21
	*MYO7A*	1.63	0.66	1.32	0.84	0.44
adhesion	*CDH1*	1.12	0.92	1.29	1.02	0.81
	*CDH11*	1.35	0.86	1.02	0.44	1.25
	*CDH15*	1.31	1.64	0.58	1.71	0.85
	*CDH2*	1.29	1.07	0.92	0.55	0.56
	*CDH22*	5.32	0.42	0.52	0.80	0.71
	*CDH26*	1.61	1.19	0.90	1.43	0.90
	*CDH4*	0.86	1.01	0.93	0.56	0.63
	*CDHR2*	1.40	2.25	1.83	0.67	0.81
	*CDHR5*	1.00	1.38	1.10	0.67	0.42
	*CLEC16A*	1.37	1.02	1.22	0.78	0.76
	*CTNNA2*	1.24	1.00	0.82	1.40	0.39
	*CTNNB1*	1.44	1.08	1.34	0.81	0.75
	*DSP*	1.36	1.17	1.50	0.77	1.18
	*GJA1*	1.94	0.55	0.76	0.67	0.43
	*GSC*	0.87	1.21	1.30	0.93	1.24
	*ICAM1*	1.30	0.98	1.47	0.79	0.80
	*ICAM5*	1.00	1.21	1.55	0.79	0.70
	*MUC1*	0.86	0.81	1.22	1.07	1.13
	*NID1*	1.32	0.73	0.79	0.48	0.41
	*OCLN*	1.30	1.49	1.88	0.56	0.71
	*PLEC*	1.25	1.06	1.10	0.76	0.59
	*S100A4*	1.33	0.77	0.95	0.85	1.52
	*SDC1*	1.95	0.66	0.86	0.87	2.34
	*SERPINH1*	1.13	0.91	1.07	1.00	0.66
	*TJP1*	1.23	1.12	1.55	0.71	0.41
	*TJP3*	1.03	0.69	1.06	1.02	0.95
cell polarity	*LIN7A*	1.28	1.23	1.22	0.85	0.61
ECM	*COCH*	1.42	0.60	1.25	0.80	0.31
	*COL14A1*	1.16	0.62	0.98	0.43	0.40
	*COL4A1*	1.17	0.74	0.95	0.49	0.44
	*COL4A4*	1.30	0.72	0.71	0.46	0.66
	*COL6A2*	1.09	0.64	0.65	0.85	0.72
	*COL8A1*	1.40	0.91	1.16	0.26	1.16
	*DDR2*	2.55	0.88	1.40	0.28	0.47
	*FBN1*	1.51	0.68	0.99	0.42	0.66
	*FRAS1*	2.10	1.06	1.85	0.61	1.11
	*GDF5*	0.75	0.60	0.53	0.94	0.35
	*ITGA1*	1.64	0.64	1.03	0.42	0.42
	*ITGA10*	0.88	1.39	1.31	0.76	0.91
	*ITGA5*	0.89	0.58	0.84	0.72	0.52
	*ITGA6*	1.65	0.83	1.38	0.75	0.78
	*ITGA8*	1.50	1.06	1.02	0.36	0.89
	*ITGAV*	1.58	0.91	1.54	0.70	0.68
	*ITGB2*	1.71	0.81	0.83	0.61	1.52
	*ITGB6*	4.02	0.59	1.11	0.36	0.65
	*ITGB8*	1.27	0.85	1.06	0.55	0.75
	*LAMA1*	1.13	0.88	0.68	0.91	3.43
	*LAMA2*	1.40	0.87	0.85	0.49	1.06
	*LAMA3*	1.73	0.54	1.07	0.68	0.44
	*LAMA4*	1.24	0.75	0.89	0.53	0.46
	*LAMA5*	1.20	0.76	1.14	0.86	0.58
	*LAMC2*	1.37	0.42	2.12	0.77	0.31
	*LAMC3*	1.22	0.88	0.78	0.64	1.68
	*LGALS3*	2.25	0.51	0.88	1.10	0.97
	*MATN1*	1.08	0.90	1.06	0.89	0.59
	*MATN2*	1.44	0.66	0.90	0.54	0.63
	*MMP2*	1.06	0.57	0.87	0.59	0.72
	*MMP3*	1.80	1.09	0.82	0.64	1.04
	*MMP7*	1.31	0.88	0.79	0.58	0.99
	*MMP9*	1.05	0.79	0.72	0.63	1.72
	*NPNT*	1.38	1.12	0.97	0.79	1.26
	*NTN4*	2.22	0.79	1.09	0.68	0.82
	*TNN*	1.66	0.71	0.59	0.52	1.75
	*TNXB*	1.23	0.60	0.83	0.59	0.70
	*VTN*	0.65	0.98	0.88	0.77	0.63
intermediate filaments	*FLG*	1.58	0.65	0.93	0.43	0.19
	*KRT18*	0.89	0.86	0.80	1.44	0.94
	*KRT19*	1.57	0.48	0.88	1.05	1.92
	*KRT5*	2.90	0.47	0.82	0.84	0.77
	*KRT8*	0.96	0.89	0.85	1.42	1.03
	*VIM*	1.14	0.79	0.99	0.74	0.69
microtubules	*MAP2*	1.31	1.20	1.23	0.75	0.74
	*MAP7*	1.03	0.99	1.61	0.85	0.64
	*MAPRE3*	0.95	1.02	1.17	0.97	0.46
	*MZT1*	0.94	1.11	1.13	0.78	0.63
	*TUBA4A*	1.02	0.53	0.81	1.03	0.74
	*TUBGCP4*	1.49	1.05	1.27	0.73	0.64
	*TUBGCP5*	1.30	1.11	1.21	0.90	0.76
molecular motors	*DYNC1H1*	1.32	1.14	1.27	0.88	0.66
	*KIF1B*	1.62	1.02	1.58	0.85	0.60
	*KIF21A*	1.73	1.10	1.91	0.68	0.76
	*KIF21B*	1.41	1.17	1.28	0.58	0.82
	*KIF26A*	1.48	1.30	1.11	0.54	0.77
	*KIF2A*	1.52	1.06	1.53	0.62	0.66
	*KIF3A*	1.27	1.08	1.48	0.69	0.78
	*KIF3B*	1.29	1.18	1.25	0.87	0.78
	*KIF5A*	1.33	0.67	1.33	0.52	0.25
	*KIFAP3*	1.27	1.00	1.18	0.83	0.76
TF	*FOXC2*	0.78	0.81	0.75	0.44	0.38
	*LEF1*	1.05	0.89	0.63	0.79	0.45
	*SMAD2*	1.19	0.90	1.30	0.84	0.54
	*SMAD3*	1.28	0.95	1.20	0.75	0.79
	*SNAI1*	1.08	0.63	1.02	0.73	0.62
	*SNAI2*	2.65	0.60	0.71	0.72	0.47
	*TCF3*	1.10	1.00	1.29	0.89	0.79
	*TCF4*	1.56	0.76	1.30	0.60	0.72
	*TWIST1*	0.78	0.77	0.98	1.23	1.20
	*ZEB1*	1.31	0.75	1.06	0.55	0.41
	*ZEB2*	1.53	0.82	1.07	0.43	0.87

**Table 7 ijms-24-00164-t007:** Detailed characteristics of the studied cohort.

	All (n = 397)	≤55 Years Old (n = 109)		60–70 Years Old (n = 253)		70> Years Old (n = 35)	
		Early (n = 53)	Late (n = 56)	Early (n = 84)	Late (n = 169)	Early (n = 13)	Late (n = 22)
Age	63 (41–78)	51	53	64	65	72	72
-median							
Race							
-Asian	2	-	1	-	1	-	-
-Black or African American	6	1	2	-	2	1	-
-White	125	15	18	32	49	7	4
-NA	264	37	35	52	117	5	18
Tumor status	250	41	37	63	88	11	10
-tumor free	65	3	12	3	41	-	6
-with tumor	82	9	7	18	40	2	6
-NA							
Vital status	391	53	54	83	166	13	22
-alive	6	-	2	1	3	-	-
-dead							
Adjuvant radiation treatment	191	30	23	42	79	5	12
-no	32	-	5	1	26	-	-
-yes	174	23	28	31	64	8	10
-NA							
Treatment outcome of first course	124	18	14	30	49	5	8
-complete remission/response	23	-	4	-	14	-	4
-partial remission/response	19	-	-	2	17	-	-
-stable disease	12	-	4	-	8	-	-
-progressive disease	219	35	34	50	81	8	10
-NA							
Laterality	343	43	52	71	146	11	20
-bilateral	17	3	2	1	10	1	-
-left	31	6	1	12	9	1	2
-right	6	1	1	-	4	-	-
-NA							
Residual tumor	254	43	28	71	92	10	10
-R0	117	7	22	7	68	2	11
-R1	4	1	-	1	1	-	1
-R2	12	-	2	3	7	-	-
-RX	10	2	4	2	1	1	-
-NA							
New tumor event	195	34	25	48	73	5	10
-no	53	-	9	4	37	-	3
-yes	148	19	22	32	59	8	9
-NA							
Targeted molecular therapy			21	43	84	5	10
-no	193	30	6	-	21	-	2
-yes	29	-	29	41	64	8	10
-NA	175	23					
Stage							
T [clinical/pathologic]							
-T1a	1/-	-	-	-	1/-	-	-
-T1b	1/-	-	-	-	1/-	-	-
-T1c	141/-	30/-	17/-	45/-	43/-	3/-	3/-
-T2	11/-	2/-	1/-	2/-	3/-	-	3/-
-T2a	44/12	6/4	7/-	9/6	18/-	1/2	3/-
-T2b	45/11	3/4	9/-	7/6	22/-	1/1	3/-
-T2c	37/128	6/45	4/2	8/71	18/-	1/10	-
-T3a	28/127	-	7/28	2/-	15/91	-	4/8
-T3b	14/110	-	3/25	-	10/73	-	1/12
-T4	8-Feb	-	1/1	-	1/5	-	-/2
-NA	73/1	6/-	7/-	11/1	37/-	7/-	5/-
N [pathologic]							
-N0	282	35	34	68	121	10	14
-N1	60	-	16	-	37	-	7
-NA	55	18	6	16	11	3	1
M [clinical]							
-M0	362	46	53	80	152	10	21
-M1a	1	-	-	-	1	-	-
-M1c	1	-	1	-	-	-	-
-NA	33	7	2	4	16	3	1
Zone of origin							
-central zone	4	1	-	1	2	-	-
-overlapping/multiple zones	98	7	13	10	55	1	12
-peripheral zone	112	15	13	25	52	3	4
-transition zone	6	1	3	1	-	-	1
-NA	177	29	27	47	60	9	5
Gleason score							
-6	33	14	1	8	6	3	1
-7	203	35	29	63	62	7	7
-8	49	4	8	6	26	3	2
-9	109	-	17	7	73	-	12
-10	3	-	1	-	2	-	-
Primary pattern							
-2	1	-	-	-	-	-	1
-3	149	44	14	50	29	8	4
-4	208	9	34	33	116	5	11
-5	39	-	8	1	24	-	6
Secondary pattern							
-3	127	21	18	30	50	5	3
-4	185	30	26	47	61	8	13
-5	85	2	12	7	58	-	6
Biochemical recurrence indicator							
-no	295	43	41	68	114	10	19
-yes	46	1	8	4	33	-	-
-NA	56	9	7	12	22	3	3

## Data Availability

The datasets used for this study can be found in the Gene Expression Omnibus (accession number: GSE62944 and GSE141551; https://www.ncbi.nlm.nih.gov/geo/; accessed on 10 March 2021). Appendix A supporting the presented findings are available in the GitHub repository (https://github.com/orzechmag/notch-age-pc).

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
