# Peer review of "Age- and Stage-Dependent Prostate Cancer Aggressiveness Associated with Differential Notch Signaling"

_ijms, 2022, doi:10.3390/ijms24010164_

Round 1

Reviewer 1 Report

In the manuscript by Orzechowska et al, the authors performed  a comprehensive bioinformatics analysis  to investigate the prognostic values of individual member of the Notch signaling pathway in PC progression, according to age of cancer onset and tumor stage. 

The main read out of this study is that the core Notch signaling shows expression signatures that are age- and tumor stage specific and may predict unfavourable outcomes. Also, functional network analysis suggest that Notch signaling unfavourable signature are associated with a Notch-EMT regulatory loop, including  kinesin motors, integrin signaling, and cellular migration.

In this study, the bioinformatic analysis of the data provided by TGCA was conducted accurately and contains new information that may well benefit PC diagnostics. As regard the text concerning the description and interpretation of the results, I believe it should be more concise and less speculative, as it risks confusing rather than clarifying the data that are well depicted in the various figures and tables. 

Author Response

Dear Reviewer,

We appreciate the time and effort you have dedicated to providing valuable feedback on our manuscript. We thank you for giving us the opportunity to submit a revised draft of our manuscript titled: ”Age- and stage-dependent prostate cancer aggressiveness associated with differential Notch signaling”. We hope that the provided amendments improve the manuscript accordingly. We modified the descriptions into a more concise and less speculative form, however, as this is a subjective feeling, we are aware that each reader may view the manuscript in a different way. We also considered the opinion of the other reviewer to expand on some descriptions, thereby we hope that the changes made to the manuscript are sufficient and are a kind of consensus that suits everyone.

Sincerely yours,

Magdalena Orzechowska

Reviewer 2 Report

The authors presented a very interesting bioinformatics study that uses many modern methods in silico.

The experimental designs and results are well explained. I suggest to the authors to better characterize three main points. A) in introduction: the limitations of PSA which is not quoted in this paper as recommended biomarker for PCa. You can find relevant suggestions in this paper reporting also about the recent updates of current Guidelines concerning PSA-based diagnostic of PCA. Ferraro S, et al. Serum prostate specific antigen (PSA) testing for early detection of prostate cancer: Managing the gap between clinical and laboratory practice.  Clin Chem 2021; 67:602-9.

The GO enrichment, the pathway and Protein-Protein interaction analysis, are not commented or described and some text in figures are unreadable. A concise outcome related to the contrast in consideration (tumor against normal prostate samples) should be described in the main text, supported by a few main figures and the ones not essential should be included in the supplementary. Only the PPI hub genes are actually used in the downstream analysis, a more detailed description of them would be beneficial for the clarity of the text.

Functional characterization (gene ontology enrichment analysis) of the target genes in the network would be beneficial for the completeness of the study.

Method should include more details, eg. what method was used for clustering in 2.4; how KEGG analysis and visualization was performed; what are the configurations in PPI analysis.

Author Response

Dear Reviewer,

We appreciate the time and effort you have dedicated to providing valuable feedback on our manuscript. We thank you for giving us the opportunity to submit a revised draft of our manuscript titled: ”Age- and stage-dependent prostate cancer aggressiveness associated with differential Notch signaling”.

Comment 1: The experimental designs and results are well explained. I suggest to the authors to better characterize three main points. A) in the introduction: the limitations of PSA which is not quoted in this paper as recommended biomarker for PCa. You can find relevant suggestions in this paper reporting also about the recent updates of current Guidelines concerning PSA-based diagnostic of PCA. Ferraro S, et al. Serum prostate specific antigen (PSA) testing for early detection of prostate cancer: Managing the gap between clinical and laboratory practice.  Clin Chem 2021; 67:602-9.

Response: We agree that doubts on the clinical usage of PSA should be highlighted, therefore we added a short paragraph based upon the review of Ferraro et al.

Comment 2:The GO enrichment, the pathway and Protein-Protein interaction analysis, are not commented or described and some text in figures are unreadable.

Response: The present study is based on the a priori-defined biological and functional groups of genes. Besides the technical description of the analyses performed included in the Materials and Methods section, we explained each step of the work in the Results enabling the reader to follow the content. The biological annotation of the studied genes was performed according to the general knowledge as well as our previous studies enabling to split of the genes according to their functions in a cell. In the present manuscript, there was thus no GO enrichment performed per se other than the corroboration of the findings with the literature and molecular biology in general. We admit that it was not clearly highlighted in the Materials and Methods and we added a proper description. All of the results shown are described and commented on in the text. The PPI networks have been discussed in section 3.3. We admit that some of the figures are less readable probably due to embedding them into a text. As original files, these figures are of high quality. Nevertheless, we apologize for that and refer you to evaluate the original figures in a stand-alone version.

Comment 3: A concise outcome related to the contrast in consideration (tumor against normal prostate samples) should be described in the main text, supported by a few main figures and the ones not essential should be included in the supplementary. Only the PPI hub genes are actually used in the downstream analysis, a more detailed description of them would be beneficial for the clarity of the text.

Response: In the study, we employed TCGA data split into subgroups of particular ages and stages of the disease. We, therefore, showed the contrasts between these subgroups that varied in clinical outcomes, as shown in the Kaplan-Meier curves. The demonstrated results are a biological elucidation of why some individuals have worse outcomes than expected, i.e. we showed a molecular reflection underlying the observed phenotype among PC patients. We focused on the major processes associated with disease progression, which has been in our opinion sufficiently described in both Results and Discussion sections. The full version of the results has been moved to the Supplements to avoid overloading the main manuscript. Furthermore, we neither employed nor analyzed the contrasts between normal and tumor tissues as it was impossible to perform due to a lack of matched normal samples for comparison. Importantly, our results were validated with an independent cohort which resembled the primary data in its design.

Comment 4: Functional characterization (gene ontology enrichment analysis) of the target genes in the network would be beneficial for the completeness of the study.

We already performed a general GO enrichment analysis of the downstream targets of the Notch signaling pathway, which results were shown in a previous study [doi: 10.3389/fcell.2020.592616; doi: 10.3390/cancers13040768; doi: 10.1371/journal.pone.0188842]. We provided a proper reference within the text. In the present study, we described the relevant genes and as it was aforementioned did not perform any enrichment, because these genes have been already functionally characterized. More specifically, we did not use differentially expressed genes, but the genes selected by their functional relevance annotated via e.g. KEGG database and the current literature. The whole study focused actually only on the functional characteristics providing the systems biology-based global insight into a cell as a whole.

Comment 5: Method should include more details, eg. what method was used for clustering in 2.4; how KEGG analysis and visualization was performed; what are the configurations in PPI analysis.

We added details that were appropriate and applicable. We hope that the provided amendments improve the manuscript accordingly.

Sincerely yours,

Magdalena Orzechowska

Round 2

Reviewer 2 Report

The authors answered all of my concerns.